# Epigenomic charting and functional annotation of risk loci in renal cell carcinoma

**Amin H. Nassar** [1,2,3,12], **Sarah Abou Alaiwi**[2,3,4,12], **Sylvan C. Baca**[2,3,12], **Elio Adib**[2,3], **Rosario I. Corona** [5,6,7], **Ji-Heui Seo**[3], **Marcos A. S. Fonseca**[5,6], **Sandor Spisak** [3,4,8], **Talal El Zarif**[2,3], **Viktoria Tisza**[3,4,8], **David A. Braun** [1,3,8], **Heng Du** [2], **Monica He**[3], **Abdallah Flaifel**[9], **Michel Alchoueiry** [2], **Thomas Denize**[9], **Sayed G. Matar**[9], **Andres Acosta**[9], **Sachet Shukla** [3,10], **Yue Hou**[3,10], **John Steinharter**[3], **Gabrielle Bouchard**[3], **Jacob E. Berchuck** [2,3], **Edward O'Connor**[3], **Connor Bell** [3], **Pier Vitale Nuzzo** [3], **Gwo-Shu Mary Lee**[3], **Sabina Signoretti**[9], **Michelle S. Hirsch** [9], **Mark Pomerantz**[3], **Elizabeth Henske** [2,3], **Alexander Gusev** [3,11], **Kate Lawrenson** [5,6,7], **Toni K. Choueiri** [2,3] ✉, **David J. Kwiatkowski** [2,3] ✉ & **Matthew L. Freedman** [3,4,8] ✉

While the mutational and transcriptional landscapes of renal cell carcinoma (RCC) are well-known, the epigenome is poorly understood. We characterize the epigenome of clear cell (ccRCC), papillary (pRCC), and chromophobe RCC (chRCC) by using ChIP-seq, ATAC-Seq, RNA-seq, and SNP arrays. We integrate 153 individual data sets from 42 patients and nominate 50 histology-specific master transcription factors (MTF) to define RCC histologic subtypes, including *EPAS1* and *ETS-1* in ccRCC, *HNF1B* in pRCC, and *FOXI1* in chRCC. We confirm histology-specific MTFs via immunohistochemistry including a ccRCC-specific TF, BHLHE41. FOXI1 overexpression with knock-down of EPAS1 in the 786-O ccRCC cell line induces transcriptional upregulation of chRCC-specific genes, *TFCP2L1, ATP6VOD2, KIT*, and *INSRR*, implicating FOXI1 as a MTF for chRCC. Integrating RCC GWAS risk SNPs with H3K27ac ChIP-seq and ATAC-seq data reveals that risk-variants are significantly enriched in allelically-imbalanced peaks. This epigenomic atlas in primary human samples provides a resource for future investigation.

In 2021, an estimated 76,080 adults in the United States will be diagnosed with kidney cancer, and an estimated 13,780 deaths from this disease will occur[1]. Renal cell carcinomas (RCCs), the most common family of kidney tumors and one of the top ten most common cancers in the US, are further stratified into several histologic subtypes. The most common subtype is clear cell RCC (ccRCC), accounting for ~75% of cases; papillary RCC (pRCC) and chromophobe RCC (chRCC) account for ~15% and 5% of cases, respectively[2]. These subtypes display divergent clinical behavior with regard to prognosis and response to therapeutic agents[3–5]. Large-scale molecular profiling efforts have characterized the genomic and transcriptomic landscapes of ccRCC[6], pRCC[7], and chRCC[8]. These analyses revealed remarkable heterogeneity among these forms of RCC, with each subtype exhibiting distinctive somatic mutations, chromosomal copy number alterations, and gene expression profiles[3]. Notably, the histone modifications and sites of chromatin accessibility driving the transcriptional landscapes of RCC histotypes, and how this relates to kidney cancer heritability, have not been systematically explored.

Enhancers are *cis*-acting DNA regions that bind *trans*-acting proteins to determine cell-type-specific gene expression patterns and

responses to internal and external signals[9,10]. Chromatin immunoprecipitation and sequencing (ChIP-seq) of post-translational histone modifications (e.g., H3K27ac and H3K4me2) have identified millions of enhancers in mammalian genomes with the number of active enhancers in any given cell type estimated to be in the tens of thousands[10,11]. Profiling enhancers has emerged as a powerful tool to characterize critical transcription factors (TFs) driving cellular transcriptional states and to better understand germline risk variants. It has been established that cellular identity and function is determined primarily by a subset of TFs termed "master" TFs[12–16]. These TFs occupy active enhancers in the cell, and preferentially bind within exceptionally large enhancer domains termed "super-enhancers" (SEs) or stretch enhancers, that regulate genes required for establishing cell identity and function[17,18]. Moreover, master TFs participate in interconnected auto-regulatory circuitries or "cliques" that are self-reinforcing, show marked cell selectivity, and function to maintain cell state and/or cell survival[17,19]. In addition, analyses of population-based epigenomes have further revealed that expression quantitative trait loci (eQTLs) SNPs are often also associated with variation in nearby epigenomic features (such as active enhancers marked by H3K27ac) in coordinated regulatory modules[20–25], motivating the use of epigenetic datasets for better functional characterization of these loci. Many of the prior epigenetic data in RCC were based on cell lines, which diverge from their original tumors and do not represent all histologic subtypes[26].

Herein, we define the epigenetic architecture and circuitry of RCC across different histologic subtypes in 42 primary human RCC tumors using a combination of genome-wide H3K27ac, H3K4me2, and chromatin accessibility assays, and integrate data from targeted DNA sequencing and bulk RNA sequencing. We delineate distinctive enhancers operative in the different RCC histologies, nominate putative histology-specific master TFs, and prioritize RCC GWAS risk loci for functional validation.

## Results

### Mapping the chromatin regulatory landscape of RCC

To examine the cistrome and to identify master TFs across the histologic subtypes of RCC, we performed histone chromatin immunoprecipitation followed by sequencing (ChIP-seq), the Assay for Transposase-Accessible Chromatin using sequencing (ATAC-seq), and RNA sequencing (RNA-seq) on 42 fresh frozen RCC tumor samples (24 ccRCC, 6 pRCC, 12 chRCC). Thirty-eight of 40 (95%) tissues were derived from radical nephrectomies. Of the 6 pRCC tumors, 4 were type I, 1 was type II, and 1 was unknown. Patients were grouped into two cohorts (1 and 2) as defined in Supplementary Data 1, Supplementary Fig. S1, and the Results section. We conducted H3K4me2 ChIP-seq to map both active and poised enhancers[27] and H3K27ac ChIP-seq to identify active promoters and enhancers[28]. ATAC-seq was performed to define the chromatin accessibility landscape, and RNA-seq was performed to capture the transcriptional programs of each RCC subtype. A total of 153 libraries were generated across the different datatypes (Supplementary Data 1 and 2 and Supplementary Fig. S1). Using cohort 1 to assess the regulatory landscape across the different RCC histologies, a total of 153,321 promoter-distal (enhancer) H3K27ac ChIP-Seq peaks were identified across all the samples, most of which ($n = 136,469$) were common to two or more histologies (Fig. 1A, B). For example, the *PAX8* locus is marked by H3K27ac in all samples. *PAX8* is a TF involved in early kidney embryogenesis and oncogenesis in RCC[29,30] and is a clinical diagnostic tool to help differentiate RCC from other malignancies[31] (Fig. 1C). Unsupervised hierarchical clustering (Fig. 1D) and principal component analysis (PCA) (Supplementary Fig. S2A) of H3K27ac peaks clearly segregated the three histologic types of RCC, and both analyses demonstrated that the H3K27ac landscape in chRCC was more distinct than either pRCC or ccRCC. Moreover, unsupervised hierarchical clustering of our cohort with an independent cohort of matched normal ($n = 10$) from the KIRC TCGA cohort[32] showed a clean

separation of normal tissue from the different tumor histotypes (chRCC, ccRCC, pRCC, Supplementary Fig. S2B, S2C). In total, 16,852 peaks were significantly increased or decreased in one tumor histology compared to the other two (e.g., chromophobe versus clear cell and papillary) ("Methods", false discovery rate (FDR) of 0.001 and least a fourfold difference in mean peak intensity between groups). In all, 12,908 sites were upregulated in one histology: 8939 were chRCC-specific, 3653 were pRCC-specific, and 316 were ccRCC-specific (Fig. 1E). These histology-specific H3K27ac peaks were differentially marked by H3K4me2 ChIP-Seq and were associated with open chromatin, strongly suggesting that they were histology-specific active enhancers (Supplementary Fig. S2D). Moreover, differential epigenetic sites correlated with the nearest gene expression difference ($P$ value $<2 \times 10^{-16}$, chi-square test) (Fig. 1F–H). GREAT[33] analysis of the 8939 chRCC-specific H3K27ac peaks revealed enrichment for genes involved in actin regulation, fatty acid oxidation, and ion transmembrane transporter activity (Fig. 2A), consistent with previously reported mRNA signature analysis showing an increased ion transmembrane transport signature in chRCC[34]. A similar analysis of the 3653 pRCC-specific sites showed enrichment for genes involved in renal system development (Fig. 2B). This is consistent with the notion that pRCC arises from embryonic nephrogenic rest precursor lesions, which persist during adult life[35]. Since there was a relatively small number of ccRCC-specific sites ($n = 316$) in comparison to chRCC and pRCC, and the enhancer landscape of ccRCC resembled that of pRCC more than chRCC, we compared H3K27ac sites between ccRCC and pRCC only. The majority of H3K27ac sites were common to the two histologies ($n = 113,786$), while 1265 sites were ccRCC-specific, and 2661 sites were pRCC-specific (Supplementary Figs. S2E and S3A). The 1265 ccRCC-enriched peaks were associated with genes involved in circulatory system development and angiogenesis (Fig. 2C), while the 2661 pRCC-enriched peak genes were again enriched for genes involved in kidney embryogenesis (Supplementary Fig. S3B). Similar analysis of the H3K4me2 peaks demonstrated clear separation of chRCC from the other two RCC subtypes with comparable histology-specific and common "poised" sites (Supplementary Fig. S3C–3F). H3K27ac and H3K4me2 ChIP-seq signals for all samples were strongly correlated (Pearson correlation, $r = 0.73$) (Fig. 2D).

De novo motif analysis of H3K27ac peaks enriched in each subtype identified four motifs that were highly enriched in chRCC, including one resembling a forkhead motif, and another resembling the motif associated with TFCP2 (Fig. 2E). FOXI1, a forkhead family TF, and TFCP2L1, closely related to TFCP2, have both been implicated in the development of intercalated cells of the kidney, the putative cell of origin of chRCC[36]. *FOXI1* and *TFCP2L1* gene loci were characterized by high H3K27ac signal in chRCC compared to the absent or markedly lower signal in ccRCC and pRCC (Fig. 2F, G). Motif enrichment analysis of putative ccRCC-specific enhancers identified a motif resembling the basic Leucine Zipper (bZIP) BATF motif, and another resembling the basic helix-loop-helix (bHLH) TF family member HIF2α (also known as EPAS1) (Fig. 2H). *ETS1*, in the bZIP family with BATF, has been implicated in von-Hippel Lindau (VHL)-dependent ccRCC tumorigenesis[37], and was highly marked by H3K27ac in ccRCC, less so in pRCC, and not in chRCC (Fig. 2I). *EPAS1* is well-known to be dysregulated in ccRCC due to the loss of the *VHL* protein function. It is the main driver of ccRCC and is upstream of multiple critical oncogenic pathways. Recent clinical trials with HIF2 inhibitors have shown clinical activity in advanced ccRCC patients[38,39]. The top-scoring pRCC-specific motif corresponded to HNF1β (Fig. 2J, K). *HNF1β* is a member of the homeodomain-containing superfamily of TFs, which is involved in nephrogenesis[40], and was highly marked by H3K27ac specifically in pRCC (Fig. 2K).

RCC has recurrent mutations in genes involved in the chromatin remodeling and histone methylation pathways[7,8,41–44]. In our dataset, loss-of-function single-nucleotide variant and copy number alterations

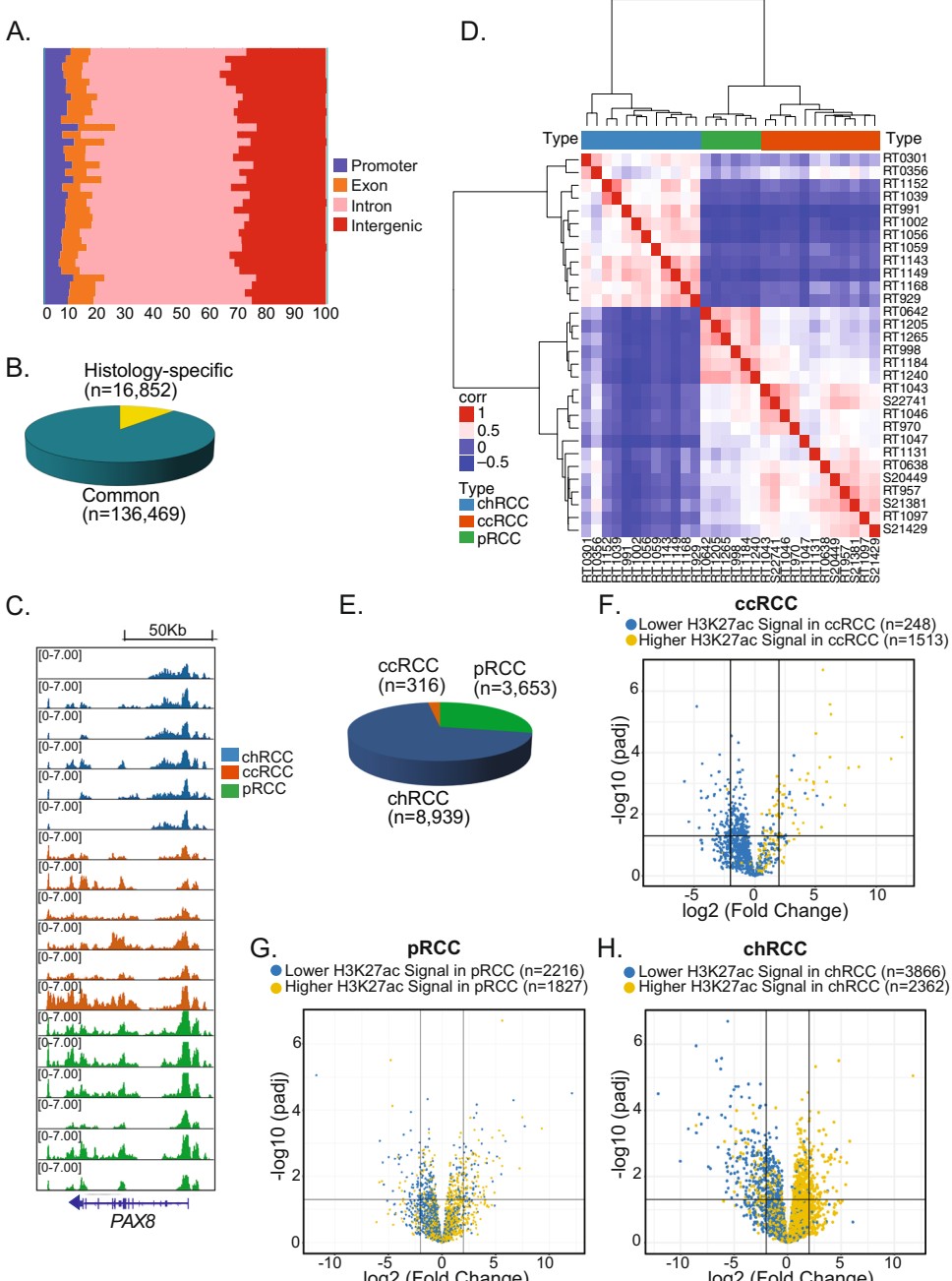

**Fig. 1 | Landscape of H3K27ac signals across RCC histologies. A** Distribution of RCC H3K27ac peaks according to genomic region for 30 fresh frozen RCC tumor samples (12 chRCC, 6 pRCC, 12 ccRCC). **B** Numbers of histology-specific and common H3K27ac peaks. **C** H3K27ac profiles at PAX8 in six representative samples from each RCC histology. **D** Hierarchical clustering of chRCC, ccRCC, and pRCC based on sample-to-sample pairwise correlation of the H3K27ac ChIP-seq peaks.

**E** Distribution of histology-specific H3K27ac peaks among RCC subtypes. **F**–**H** Volcano plots with the log change of gene expression (FPKM) in one histology compared to the other two histologies (**F** ccRCC vs. others, **G** pRCC vs. others, **H** chRCC vs. others). Two-sided *P* values were used and corrected for multiple comparison testing (FDR-adjusted *P* value <0.05). RCC renal cell carcinoma, chRCC chromophobe RCC, ccRCC clear cell RCC, pRCC papillary RCC.

(see "Methods") in such commonly mutated genes in RCC (*VHL*, *PBRM1*, *BAP1*, and *TP53)* did not correlate with global acetylation differences in both supervised and unsupervised analyses, albeit the small sizes were small. (Supplementary Fig. S4A–S4D and Supplementary Data 3 and 4).

### Specific master transcription factors define RCC subtypes
We next sought to systematically identify candidate histology-specific master TFs that define the three subtypes of RCC. Master TFs typically bind within SEs[17,45], are often regulated by SEs, and regulate one another in a transcriptional core regulatory circuit (CRC)[46]. We

employed an integrative approach[47,48], leveraging the RNA-Seq, ChIP-Seq, and ATAC-Seq data to identify candidate histology-specific master TFs (Fig. 3A). This approach aims to utilize orthogonal information to identify a consensus set of master TFs. We combined (1) expression data of differentially expressed TFs across RCC histologic subtypes; (2) TFs specific to RCC histologic subtypes relative to other cancer types (CaCTS)[49]; (3) differential SE-associated TFs among RCC histologic subtypes; and (4) TFs with histology-specific connectivity in regulatory cliques (see "Methods"). These four analyses identified more than 200 candidate TFs showing a histology-specific association in one or more analysis. We prioritized candidates for downstream validation by

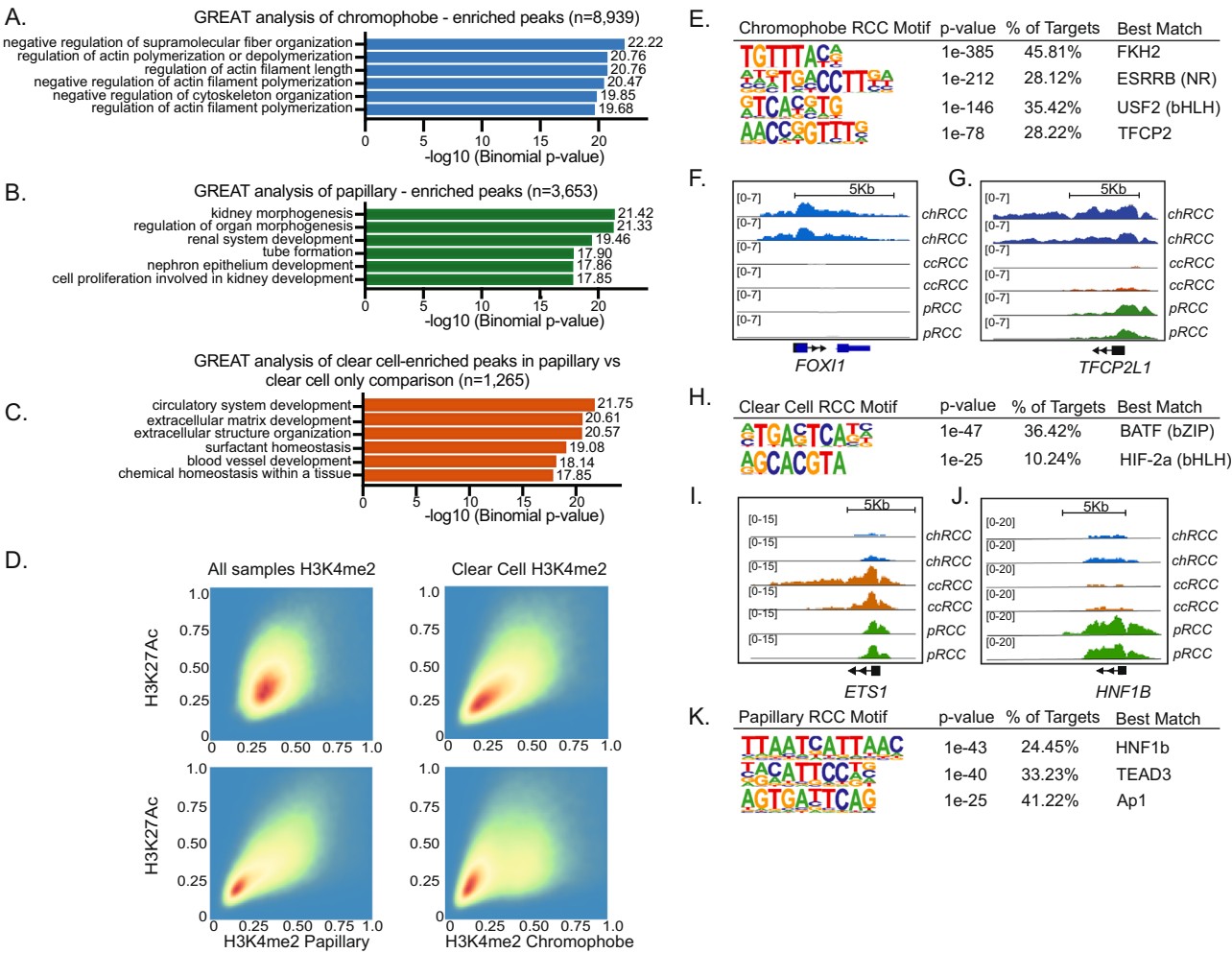

**Fig. 2 | Epigenetic annotation of regulatory elements identifies enrichment of histology-specific pathways and TFs. A** GREAT analysis of chromophobe-enriched peaks (*n* = 8939). Two-sided *P* values are shown. **B** GREAT analysis of papillary-enriched peaks (*n* = 3653). Two-sided *P* values are shown. **C** GREAT analysis of clear cell-enriched peaks in clear cell vs. papillary only comparison (*n* = 1265). **A**−**C** GREAT calculates statistical enrichments for association between genomic regions and annotations. Two-sided *P* values are shown. **D** Density map of correlation between H3K27ac versus H3K4me2 ChIP-seq peaks across subtypes. **E** Four most significantly enriched nucleotide motifs present in chRCC-specific sites by de novo motif analysis, limited by ATAC peaks. **F**, **G** H3K27ac profiles near

FOXI1 and TFCP2L1, respectively, in two representative samples of each histology (chRCC, ccRCC, pRCC). **H** Two most significantly enriched nucleotide motifs present inccRCC specific sites by de novo motif analysis. *P* values are two-sided and unadjusted for multiple comparisons. **I**, **J** H3K27ac profiles near ETS-1, and HNF1B, respectively, in two representative samples of RCC histology. **K** Three most significantly enriched nucleotide motifs present in pRCC-specific sites by de novo motif analysis. *P* values are two-sided and unadjusted for multiple comparisons. RCC renal cell carcinoma, chRCC chromophobe RCC, ccRCC clear cell RCC, pRCC papillary RCC.

selecting those that were identified in more than one analysis ("Methods", Supplementary Figs. S5A-S5J and S6 and Supplementary Data 5–11). This analysis highlighted 50 candidate histology-specific master TFs (*N* = 20, chRCC; *N* = 14, pRCC; *N* = 16, ccRCC) (Supplementary Data 12), including *FOXI1*, *TFCP2L1*, and *DMRT2* for chRCC; *EPAS1*, *ETS1*, *BARX2*, *ZNF395* for ccRCC, and *HNF1B* and *NR2F2* for pRCC (Fig. 3B). SE ranking, gene expression, and CES of the 50 histology-specific TFs selected using this meta-analysis approach clustered the samples tightly according to their respective histologies (Supplementary Fig. S6A).

To confirm whether similar patterns of TF-specificity are found in normal counterpart tissues, we turned to gene expression datasets from TCGA KICH, KIRC, and KIRP cohorts. GSEA analysis for histology-specific master TFs showed enrichment of clear cell and papillary RCC-specific transcription factors in tumors compared to normal (FDR corrected *q*-value <0.01). There was no significant enrichment of chromophobe RCC-specific transcription factors in tumors compared to normal. Of 50 master TFs, 34 (68%) had more significant expression in the tumor tissue compared to the normal counterpart (14/16 for

KIRC, 10/14 for KIRP, and 10/20 for KICH, Supplementary Fig. S6B–S6D).

Clinical correlative analyses from CheckMate cohorts (009/10/025) showed that among the 50 master TFs, high BARX2 expression significantly correlated with improved overall survival in the entire cohort of patients with ccRCC (Supplementary Data 13 and Supplementary Fig. S6E) and in the subgroup treated with the anti-PD1, nivolumab (Supplementary Data 13 and Supplementary Fig. S6F).

To provide proof of concept validation at the protein level, we investigated the expression specificity and localization of master TFs by immunohistochemistry (IHC). We selected four representative master TFs that met the following criteria: (1) at least twofold change in gene expression of the histology-specific TF by bulk RNA sequencing (from TCGA) compared to the two other RCC histotypes, (2) high-quality antibodies for IHC, and (3) TF was not previously validated and implemented on a clinical level. For the 4 TFs (BHLHE41, HNF1β, NKX6.1, and ZNF395), we found significant changes in protein expression levels across histologic subtypes (Fig. 3C). More specifically, two ccRCC-specific master TFs (BHLHE41 and ZNF395), were

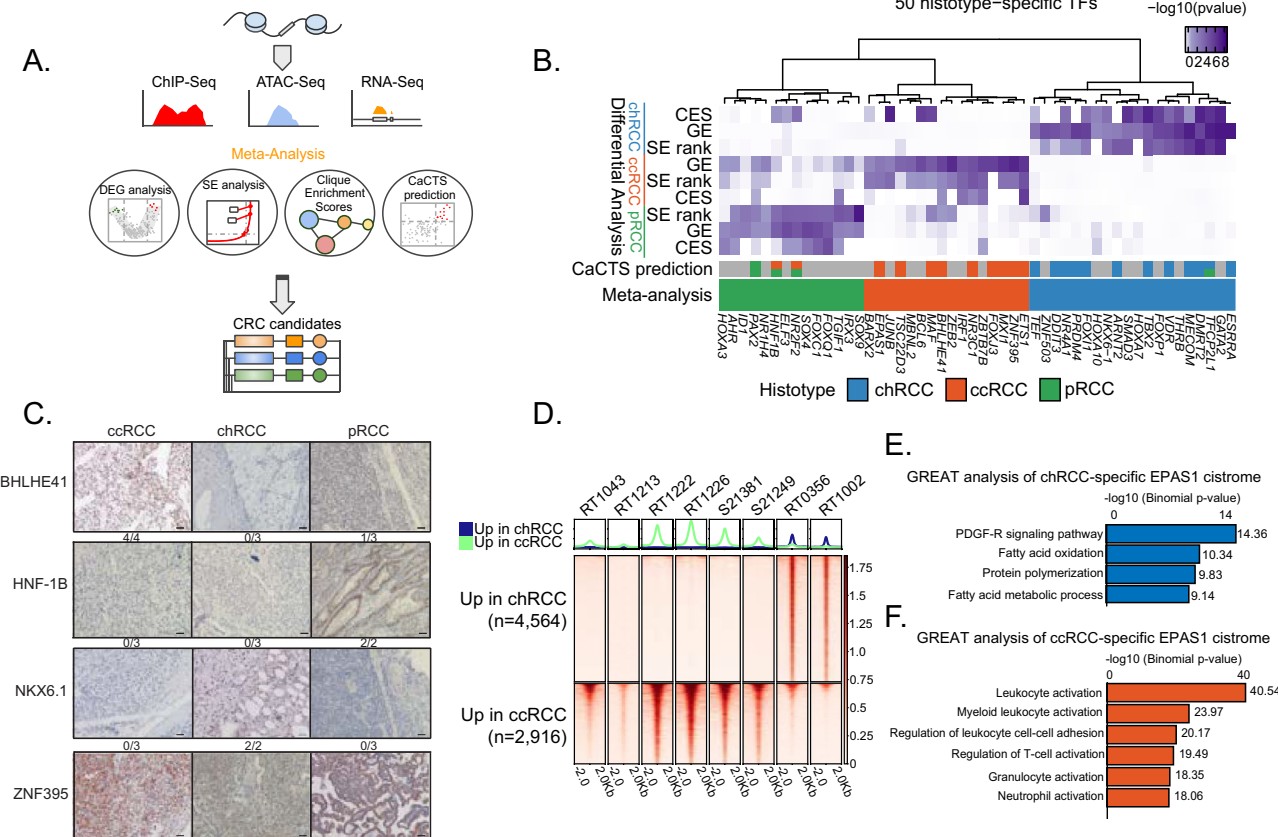

**Fig. 3 | Multi-dimensional integrative analysis identifies histology-specific master TFs. A** Overview of the approach used to nominate histology-specific master TFs participating in CRC. **B** Heatmap integrating the 50 histology-specific TFs identified by the meta-analysis approach (CES, differential expression, SE rank analysis, and CaCTS). **C** Representative immunohistochemical stainings of indicated antibodies in samples from ccRCC, chRCC, and pRCC tumors. Four histology-specific master TFs are shown. The number of positive tumors/number of tumors examined are indicated below each image (for ccRCC, $n = 4$ for TF BHLHE41, $n = 3$ for TFs HNF1B, NKX6.1, and ZNF395. For chRCC, $n = 3$ for TFs BHLHE41, HNF1B, and ZNF395, n = 2 for NKX6.1. For pRCC, $n = 3$ for BHLHE41, NKX6.1, and ZNF395, $n = 2$ for HNF1B). Scale bar is 50 μm. **D** ChIP-seq binding profiles of EPAS1 across 6 ccRCC

and 2 chRCC human tissue samples. **E** GREAT analysis of chRCC- enriched EPAS1 peaks relative to ccRCC. Two-sided $P$ values are shown. **F** GREAT analysis of ccRCC-enriched EPAS1 peaks relative to chRCC. Two-sided $P$ values are shown. **E, F** GREAT calculates statistical enrichments for the association between genomic regions and annotations. ChIP-seq chromatin immunoprecipitation sequencing, ATAC-seq assay for transposase-accessible chromatin sequencing, DEG differentially expressed genes, SE superenhancer, CaCTS cancer core transcription-factor specificity, CRC core regulatory circuitries, RCC renal cell carcinoma, chRCC chromophobe RCC, ccRCC clear cell RCC, pRCC papillary RCC. Source data are provided as a Source Data file.

---

highly expressed in ccRCC tumors. Both TFs had nuclear localization in four out of four and three out of three ccRCC samples, respectively. NKX6.1, a TF recently described as being expressed in chRCC, was detected nuclearly in two out of two chromophobe samples with no expression in either ccRCC or pRCC. We next confirmed histology-specific in vivo binding of the nominated master TF EPAS1 through examining the EPAS1 cistrome in ccRCC and chRCC. We characterized 2916 clear cell RCC-specific and 4564 chromophobe RCC- specific EPAS1-binding sites through performing EPAS1 ChIP on eight additional primary human tumors ($n = 2$ chRCC; $n = 6$ ccRCC, Fig. 3D). Subtype-specific EPAS1-binding sites were highly enriched for subtype-specific sites of H3K27ac. For instance, 2090 of the 4565 chromophobe-specific EPAS1-binding sites (46%) overlapped with chromophobe-specific H3K27ac sites ($P < 2.2e-16$). This supports the conclusion that these differential EPAS1-binding sites are non-random and biologically relevant, because they coincide with regulatory elements that segregate closely with histology. ccRCC-specific EPAS1-binding sites were enriched for immune cell and white blood cell activation pathways and chRCC-specific EPAS1-binding sites were enriched for metabolic processes and fatty acid activation (Fig. 3E, F).

To further validate our approach of the nominated master TFs in driving the transcriptional identity of the different RCC histologies, we

manipulated TF expression in a ccRCC cell line. We hypothesized that overexpression of chRCC-specific TFs and suppression of ccRCC-specific TFs can shift the transcriptional landscape of the ccRCC cell line 786-O to become more chromophobe-like. *FOXI1* scored as a chRCC-specific TF in 4/4 master TF analyses (Fig. 3B). Furthermore, FOXI1 is selectively expressed in intercalated cells (ICs), the putative cellular origin of chRCC[36,50] and is more highly expressed in chRCC than other cancer subtypes in the TCGA dataset (Supplementary Fig. S7A). We also manipulated the expression of *EPAS1* as a second target as it was highly specific for ccRCC in our integrative analysis (Fig. 3B and Supplementary Fig. S8B), and prior studies have characterized its role in the pathogenesis of ccRCC[51]. We overexpressed FOXI1 in the ccRCC cell line 786-O (FOXI1 OE); suppressed EPAS1 (EPAS1 KD) and simultaneously perturbed both genes in the same cell line (FOXI1 OE/EPAS1 KD) (Supplementary Fig. S7C–7E and Supplementary Data 14). Expression changes are described in Supplementary Fig. S8A–S8F and Supplementary Data 15–18. Gene set enrichment analysis (GSEA) of downregulated genes with FOXI1 OE/EPAS1 KD in 786-O showed enrichment of genes involved in the immune responses (Fig. 4A). Concomitant FOXI1 OE and EPAS1 KD in 786-O cells resulted in upregulation of chRCC-specific master TFs such as TFCP2L1 and NR3C2 and down-regulation of ccRCC-specific master TFs such as ETS-

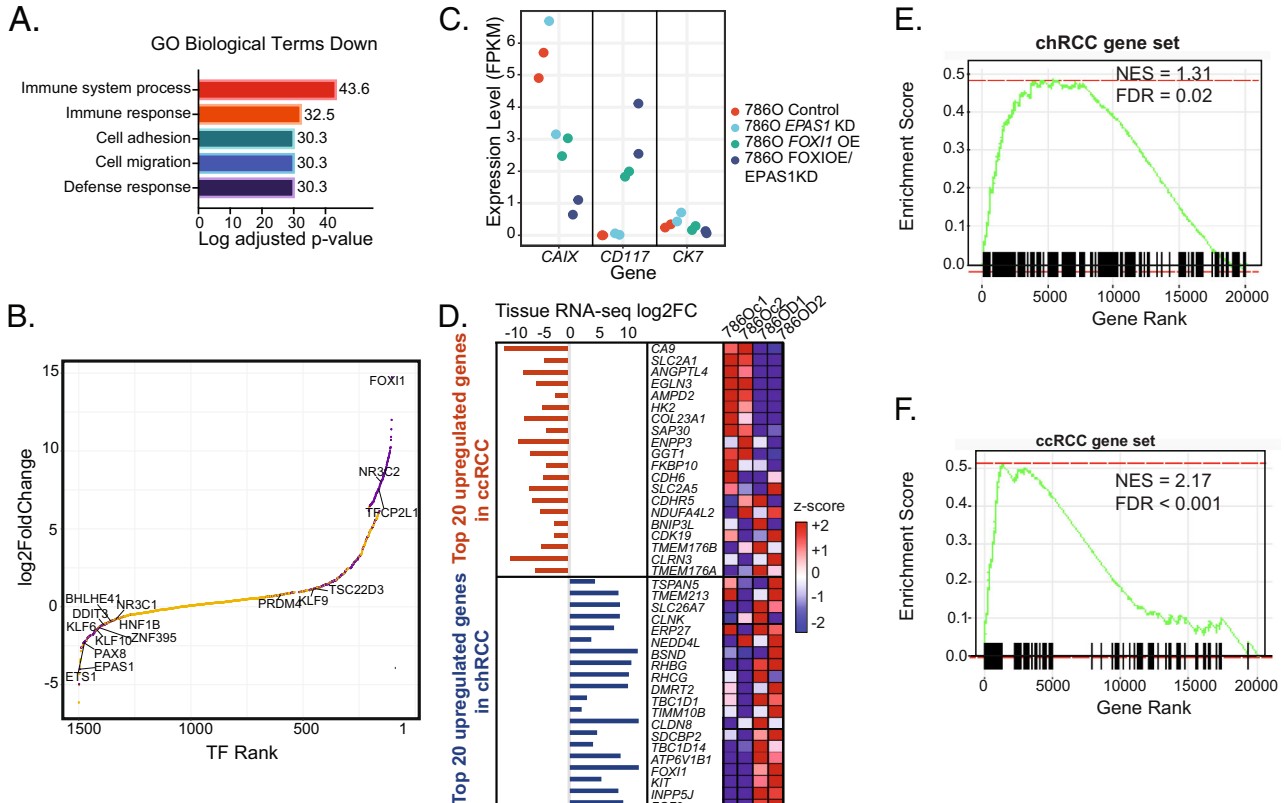

**Fig. 4 | Functional perturbation of two master TFs in a ccRCC cell line yields a more chRCC-like transcriptional profile. A** Downregulated GO biological terms in the cell line 786-O FOXI1 OE/EPAS1 KD. Two-sided adjusted *P* value corrected for multiple comparisons. **B** Rank order of differentially expressed TFs between 786-O CTRL and 786-O FOXI1 OE/EPAS1 KD by expression levels. Each TF dot represents one TF. Wald test from DESeq2 was used. Purple dots indicate adjusted *P* value <0.001 by DESeq Expression. **C** Gene expression levels of CAIX, CD117, and CK7 across different 786-O cell line conditions. **D** Heatmap showing the relative expression of the top 20 upregulated genes in ccRCC vs. chRCC and vice versa in the different cell line conditions. **E** GSEA analysis showing the top 100 upregulated genes from chRCC vs. ccRCC dataset in 786-O FOXI1 OE/EPAS1 KD vs. 786-O CTRL comparison. **F** GSEA analysis of the top 100 upregulated genes in ccRCC vs. chRCC in 786-O CTRL vs. 786-O FOXI1 OE/EPAS1 KD. CTRL control, KD knockdown, OE overexpression, TF transcription factor, GSEA gene set enrichment analysis, 786Oc1 CTRL1, 786Oc2 CTRL2, 786OD1 786-O FOXI1 OE/EPAS1 KD1, 786OD2 786-O FOXI1 OE/EPAS1 KD2, NES Normalized Enrichment Score, FDR false discovery rate, FC fold change, RCC renal cell carcinoma, chRCC chromophobe RCC, ccRCC clear cell RCC, pRCC papillary RCC.

1 and ZNF395 compared to control cells. (Fig. 4B). Further supervised analysis showed that CAIX, a downstream target of EPAS1 was significantly overexpressed in the WT 786-O cell line compared to EPAS1 KO/FOXI1 OE cell line. In contrast, CD117, a well-known immunohistochemical marker of chRCC[52], was overexpressed in the EPAS1 KO/FOXI1 OE cell line compared to WT 786-O cell line. Although CK7 expression is a supportive marker clinically for chRCC[53], its expression was uniform across WT 786-O and manipulated cell line states (Fig. 4C). Of note, there were only 109 differentially expressed genes between the FOXI1 OE/EPAS1 KD cell and FOXI1 OE only cell line (*P* < 0.05, *q* < 0.01, Supplementary Fig. S8D and Supplementary Data 17).

We then compared differentially expressed genes between our experimental conditions in 786-O and our RCC tumor expression data (Fig. 4D and Supplementary Fig. S8G). Defining chRCC and ccRCC gene sets as the top 100 upregulated genes for each histology relative to the other, we showed that the chRCC gene set is enriched among upregulated genes in 786-O FOXI1 OE/EPAS1 KD cell line vs. control (Fig. 4E), and the ccRCC gene set is enriched among upregulated genes in 786-O control vs. FOXI1 OE/EPAS1 KD cell line (Fig. 4F). The FOXI1 OE/EPAS1 KD 786-O cell line demonstrated significantly higher expression of chRCC-specific candidate master TFs, TFCP2L1, GATA2, DDIT3, NKX6-1, and lower expression of ccRCC-specific candidate master TFs, ZNF395 and TSC22D3. Differential TFs in the 786-O FOXI1 OE/EPAS1 KD vs. 786O CTRL comparison overlapped more significantly with differential TFs from the chRCC vs. ccRCC human

samples comparison than from the ccRCC vs. pRCC human sample comparison (Supplementary Data 19). In summary, these data show that overexpression of a single chRCC master transcription-factor candidate, *FOXI1*, in the ccRCC cell line 786-O, with or without knockdown of *EPAS1*, led to marked expression changes driving the cell line to be more like a chRCC cell line without any modification to the set of mutations present in 786-O cells.

**Allelic imbalance annotates germline RCC risk variants.** Allelic imbalance, the differential allelic representation of heterozygous single-nucleotide polymorphisms (SNPs) in ChIP-seq reads, provides an in vivo comparison of *cis*-regulatory activity between two haplotypes (Fig. 5A). As such, allelic imbalance can highlight the functional relevance of candidate causal variants[22,32,54–57] at loci that have been associated with RCC risk through GWAS. A key advantage of profiling epigenomes from many individuals is the ability to capture heterozygous sites and to measure the effect of TF binding on regulatory elements through analysis of allelic imbalance. To this end, we assessed chromatin allelic imbalance in ChIP-seq reads from these RCC samples in order to nominate causal risk SNPs from a genome-wide association study of RCC. We applied stratAS[32] to H3K27ac ChIP-seq data from 20 ccRCC and 6 pRCC ("Methods"). We identified 10,605 chromatin-imbalanced H3K27ac peaks—which we defined as peaks with one or more imbalanced SNPs after correction for multiple hypothesis testing—across 183,099 peaks tested in the combined ccRCC and pRCC sample sets (Fig. 5B and Supplementary Data 20).

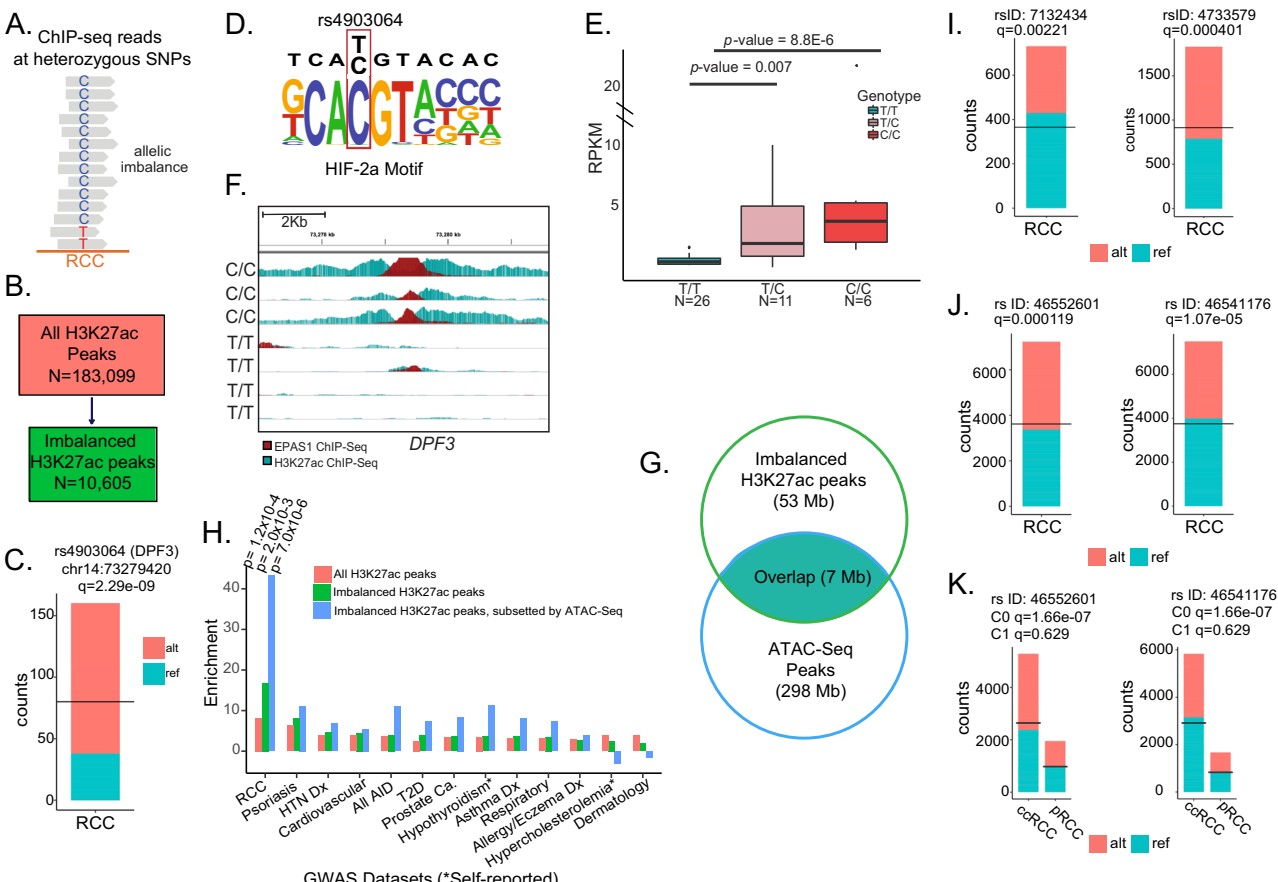

**Fig. 5 | Allelically imbalanced H3K27ac peaks in ccRCC and pRCC. A** Schematic of allelic imbalance at heterozygous SNPs. **B** Schematic showing subset of allelically imbalanced H3K27ac peaks from total H3K27ac peaks in RCC (pRCC and ccRCC) samples. **C** AI at rs4903064 (chr14:73279420; DPF3) in RCC. **D** rs4903064 C-allele sequence context creates a HIF-2a binding site. **E** RPKM values for the EPAS1 ChIP signal in the peaks around rs4903064 in 43 samples. Genotype is shown on the *X* axis. Box boundaries correspond to 1st and 3rd quartiles; whiskers extend to a maximum of 1.5× the interquartile range. Two-sided unadjusted p-values are shown. Statistical analysis was performed using Kruskal–Wallis test. **F** Overlayed EPAS1 and H3K27ac ChIP-Seq tracks for seven samples within -10 Kb of genomic coordinates of rs4903064. Individual sample genotypes are shown. **G** Venn diagram showing overlap of imbalanced H3K27ac peaks with ATAC-seq peaks in RCC. **H** Enrichment of risk SNPs from GWAS for RCC and other diseases in H3K27ac peaks with differential allelic imbalance, compared to all H3K27ac peaks. Empiric one-sided *P* value is indicated (unadjusted for multiple comparison). Statistical analysis was performed using Kruskal–Wallis test. GWAS risk SNPs rs7132434 and rs4733579 demonstrating allelic imbalance in RCC. **I** AI at Chr12 SNP rs7132423 and Chr8 SNP rs4733579 shown. **J** AI at Chr2 SNPs within EPAS1. rs46552601 and rs46541176 shown. **K** Chr2 SNPs within EPAS1, with imbalance plots split by histology. Adjusted Q values for imbalance are indicated. ChIP-seq chromatin immunoprecipitation sequencing, Ca cancer, HTN hypertension, Dx diagnosis, T2D type 2 diabetes mellitus, AID autoimmune disease, C0 reference allele, C1 alternate allele, AI allelic imbalance, RCC renal cell carcinoma, chRCC chromophobe RCC, ccRCC clear cell RCC, pRCC papillary RCC, SNP single-nucleotide polymorphism.

We hypothesized that chromatin AI peaks correspond to regulatory elements that are bound by master TFs. At these elements, the presence of *trans*-acting factors (i.e., master TFs) may result in regulatory element activity that are observable as H3K27ac AI peaks. One example is rs4903064 (chr14:73279420; 14q24.2), an eQTL for *DPF3*[58] (Fig. 5C). rs4903064, which is allelically imbalanced in our H3K27Ac dataset, has been shown to be a GWAS risk variant in all three histologic subtypes (ccRCC, chRCC, and pRCC), and the altered C-allele of this SNP has been suggested to create a HIF-binding motif (Fig. 5D)[59]. To confirm this, we used EPAS1 transcription-factor ChIP-Seq data in an independent cohort of 43 samples derived from 23 patients with ccRCC to study the effect of the GWAS risk variant rs4903064 (see "Methods") on EPAS1 binding. Analysis of 11 primary, 21 metastatic ccRCC samples, and 11 normal renal tissue samples showed that tumors with homozygous C/C alleles were significantly enriched for EPAS1 peaks (Fig. 5E, F) in both tumors and normal tissue and that rs4903064 is a *EPAS1* cQTL. This confirms prior literature that the C-allele creates a HIF-binding site[59].

We applied our chromatin AI analysis to annotate risk SNPs identified by a GWAS for RCC[58]. Chromatin AI peaks were highly enriched (16.7-fold) for RCC GWAS risk variants as assessed by LD score regression analysis ($P = 1.9 \times 10^{-4}$)[60]. Subsetting imbalanced H3K27ac peaks to regions of accessible chromatin where TFs are likely to bind (as assessed by ATAC-Seq from RCC tissues[61]) resulted in substantial additional enrichment (43.2-fold; $P = 7.0 \times 10^{-6}$, Fig. 5G, H). This enrichment represents more than fivefold that of the total set of H3K27ac peaks, which are themselves enriched 8.1-fold ($P = 1.2 \times 10^{-4}$). By contrast, multiple other GWAS phenotypes from the UK Biobank[62] showed substantially less enrichment compared to RCC GWAS SNPs, indicating the specificity of this enrichment for RCC (Fig. 5H).

Using this method, we were able to fine-map a total of 30 risk SNPs (Supplementary Data 21), with some examples highlighted. Rs7132434, located on chr 12, has been characterized as a functional variant that alters AP-1 binding leading to upregulation of *BHLHE41*, thus promoting tumor growth through induction of *IL-11*[63] (Fig. 5I). Another example is rs4733579 which is located on chr 8 and is in LD with

rs35252396 (Fig. 5I). The latter has been shown to contribute to RCC susceptibility through regulating MYC and PVT1 expression[64]. Of note, rs35252396 is an indel and therefore cannot be assessed by stratAS. Our analysis also highlighted rs46552601 and rs46541176 (Fig. 5J), both located on chr2p21 within the *EPAS1* gene, where at least 59 SNPs have been fine mapped. These two SNPs are allelically imbalanced by H3K27Ac peaks only in ccRCC but not in pRCC, consistent with the specific role of *EPAS1* (*HIF2-α*) in ccRCC pathogenesis (Fig. 5K). Further work is recommended to functionally validate the mechanism of these two SNPs in ccRCC pathogenesis. Finally, rs4765623 located in the *SCARB1* locus has been recently validated in RCC progression[32].

TFs can read the genetic code and bind *cis*-regulatory elements to activate or repress gene expression. We hypothesized that loci with significant chromatin AI are likely harbingers of TF binding and thus are associated with the regulation of gene expression in an allele-specific manner. We leveraged chromatin AI from our ccRCC H3K27Ac dataset and allele-specific expression (ASE) using RNA-seq data from the KIRC TCGA cohort. SNP loci with at least 50 reads in both the ccRCC H3K27Ac dataset and RNA-seq KIRC TCGA samples were retained. Retained SNPs were classified as chromatin allelically imbalanced or allelically balanced in ccRCC. Chromatin allelically imbalanced SNPs harbored H3K27ac peaks with a significant skew towards the alternate allele compared to the wild-type allele ($P < 0.05$). For the 22,162 chromatin allelically imbalanced SNPs, we matched a background set of chromatin allelically balanced SNPs lying within the H3K27ac consensus peak set. Allele-specific expression of genes whose transcription start site lies within 50 Kb of the chromatin allelically imbalanced and balanced SNPs was analyzed in the TCGA KIRC dataset ($N = 412$) with a significance cutoff of $P$-adjusted $<0.01$. Of the unique genes analyzed, chromatin allelically imbalanced SNPs were significantly more likely to lie within 50 kb of a gene with ASE (1170/2646, 44%) compared to the background set of chromatin allelically balanced SNPs (65/940, 6.9%, $P = 8.4e{-}25$, Supplementary Data 22–25).

## Discussion

In this study, we report a compendium of ATAC-Seq, RNA-Seq, and histone modification data across the three most common RCC subtypes. Our goal was to describe enhancer programs in primary human samples from patients with RCC and to identify histology-specific epigenetic mechanisms that may underlie differential clinical features and prognosis. Critically, this resource includes chRCC, an understudied cancer type with scarce in vitro and in vivo models and molecular profiling data, and no proven therapeutic strategies for metastatic disease.

In line with histological, transcriptional and mutational data, ccRCC, pRCC, and chRCC exhibit distinct epigenetic landscapes. chRCC clearly separated from ccRCC and pRCC, both by H3K27ac and H3K4me2 landscapes. Motif-based analysis revealed the enrichment of distinct TF binding sites in H3K27ac peaks in ccRCC vs. pRCC, suggesting that distinct TFs were active and driving histology-specific regulatory pathways and downstream transcriptional programs. Using *EPAS1* TF ChIP-Seq as an example, we demonstrate that histology-specific binding of EPAS1 mediates independent pathways in different RCC histologies.

Prior work in medulloblastoma and ependymoma demonstrated how molecularly-defined cancer subgroups exhibit specific core regulatory circuitries[65]. Herein, we nominated 50 histology-specific TFs in RCC based on a convergence of evidence, including high-level expression, specificity of expression across RCC subtypes, super-enhancer association, and connectivity. Our experimental results demonstrate that upregulating a chRCC-specific TF, FOXI1, can partially reprogram a ccRCC cell line, 786-O, towards a chRCC transcriptional phenotype; consistent with results from other tumor and normal cell types showing that master TFs govern the transcriptional identity of the cell or tissue.

Our work furthermore sheds light on multiple subtype-specific master TFs, the role of which is yet be exactly determined. *HNF1β*, a candidate pRCC master TF, has been shown to be expressed in the embryonic kidney and in pRCC, and it is upregulated in pRCC compared to other kidney cancer histologies[7,66,67]. *HNF1β* gene is located on chromosome 17 and is frequently impacted by copy number gains in pRCC[66] further providing grounds for its role in pRCC pathogenesis. ETS-1, a candidate ccRCC master TF known to have a role in cancer pathogenesis as well as angiogenesis and hematopoietic stem cell differentiation, is another example[68–71]. As a proto-oncogenic factor, *ETS-1* is capable of activating genes associated with angiogenesis, metastasis and invasive behavior in multiple tumor types[72–74]. In addition, *ETS-1* expression correlates with microvessel density in some non-glial tumors and is an independent negative prognostic marker in different tumor entities such as breast, ovarian, pancreatic, and colorectal cancers[72,75,76]. Nonetheless, the role of *ETS-1* in RCC has mainly focused on its interaction with *EPAS1* (HIF2-α)[77]. Here, we show that *ETS-1* may play a role as a master TF specific to ccRCC and not chRCC or pRCC. Future studies should focus on understanding the global epigenomic effect of *ETS1* in driving ccRCC pathogenesis.

Using clinical trial data (CheckMate 009/010/025), we suggest that high BARX2 expression may be associated with longer survival among patients treated with nivolumab but not everolimus. Prospective data in the metastatic ccRCC space are needed to confirm the role of BARX2 expression as a predictive biomarker in patients treated with immune checkpoint inhibitors.

In routine practice, pathologists use a myriad of targets to differentiate across RCC histotypes. Our approach has the potential to narrow master TFs to clinically meaningful ones and augments the current armamentarium with additional TFs, including BHLHE41 (ccRCC) and NKX6.1 (chRCC) that can be stained to better characterize tumors.

BHLHE41 was previously shown in triple-negative breast cancer to counteract expression of HIF-target genes by promoting HIF proteasomal degradation in a process independent of VHL or hypoxia[78]. Elevated protein expression of BHLHE41 in ccRCC has not been described previously. Using a systematic approach, we found histology-specific protein expression of BHLHE41 in ccRCC compared to pRCC and chRCC. In ccRCC where HIF activation is foundational, further investigation of the relation between BHLHE41 and HIF activation is warranted.

GWAS have identified hundreds of putative cancer-risk loci[79]. These studies have led to the conclusion that cancer is driven by thousands of variants with individual small effects, and that cancer-risk variants predominantly lie in non-coding regions of the genome[80,81]. More recently, it has been realized that GWAS heritability is enriched at variants that lie in tissue-relevant epigenomic features[20,60,82–84]. Emerging evidence suggests that many GWAS variants function by altering an existing or creating a TF binding site. This in turn regulates the local chromatin regulatory landscape by altering the enhancer landscape and thus modulating the expression of target genes[85]. We used our population-scale epigenomic data to further empower the discovery of *cis*-regulatory elements that may mediate between the risk variant and target gene and thus constrain putative targets for experimental validation. We showed that RCC GWAS risk SNPs are enriched in regulatory elements, consistent with prior studies[20,60,82–84]. Profiling these regulatory elements across multiple (genetically diverse) individuals allowed us to identify regions of chromatin allelic imbalance, which are further enriched for GWAS SNPs, and subsetting these regions to ATAC-seq peaks where TFs are likely to bind, showed even more enrichment for GWAS SNPs. In RCC, there are 13 recognized risk loci, but the causal variants have been identified for few, and underlying mechanisms of risk remain elusive for most regions[58]. Our approach can help pinpoint causal SNPs and provide candidate mechanisms by identifying small numbers of SNPs that may alter binding sites for

specific TFs, an example of which is the rs4903064 locus. Recent work around the rs4903064 locus[59] showed that it had an allele-specific effect on DPF3 expression in ACHN and HEK293T cell lines as assessed by massively parallel reporter assay, confirmatory luciferase assays, and eQTL analyses. The rs4903064-C RCC risk allele was shown to create a HIF-binding site and enhance gene expression. Increased expression of DPF3 conferred a growth advantage to cells by at least two pathways: inhibition of apoptosis via CEMIP and activation of STAT3 via IL23R. The authors also showed that DPF3-overexpressing cells showed higher T-cell mediated cytotoxicity compared to controls. In a separate effort, Protze et al.[86] used 23 tumor tissue specimens and two primary ccRCC cell lines and showed that the risk SNP is located within an active enhancer region, in turn creating a EPAS1-binding motif. They also showed that HIF-mediated DPF3 regulation depends on the presence of the C-risk allele. Moreover, DPF3 deletion in proximal tubular cells decreased cell growth, suggesting a role for DPF3 in cell proliferation.

Our work has several limitations. First, experimental validation was limited and performed on two master TFs in one ccRCC cell line 786-O. Future work focused on validation of the role of other candidate master TFs in lineage plasticity and across multiple RCC cell lines is warranted. Second, it is important to highlight that type 1 and type 2 papillary RCC are distinctive entities, and that although our work can globally highlight differences across RCC histotypes, the small sample size of the pRCC cohort limited more granular assessment by pRCC types.

In conclusion, this work combines and integrates ChIP-Seq, ATAC-seq, and RNA-seq data into a unified analysis to capture the deep complexity of the epigenomic landscape of RCC. The majority of epigenetic analyses have been conducted in immortalized or malignant cancer cell lines[10,48], which inevitably have diverged significantly in many respects from primary tumor tissues, as seen in prior studies from medulloblastoma and diffuse large B-cell lymphoma[26,65]. In addition, studies have demonstrated that the vast majority of available RCC cell lines are derived from ccRCC, with very few pRCC lines, and relative absence of any chRCC lines[87,88]. To date, most of our knowledge in the epigenomic space of RCC primary tissue is derived from older DNA methylation analysis[6–8], limited to one specific subtype of RCC[37], or confined to only one epigenetic dimension[61]. By integrating several epigenetic technologies, we discovered master TFs that drive histology-specific transcriptional programs in RCC and showed that regulatory elements with allelic imbalance are specifically enriched for RCC GWAS risk SNPs relative to GWAS risk SNPs for other disease features. Overall, our work highlights the histology-specific nature of these regulatory elements and the opportunity of cataloging the epigenetic landscapes in expanded patient cohorts and across various tumor types.

## Methods

### Patients and samples

All tumor samples at DFCI were obtained at the time of resection at Brigham and Women/s Hospital, and were collected under a DFCI/Harvard Cancer Center IRB-approved protocol (01-045) with the written informed consent of all patients. No monetary compensation was offered for patient participation. The samples were then stored fresh and frozen in the Gelb Center biobank at the Dana-Farber Cancer Institute, under a protocol approved by the MGB Institutional Review Board. Patients in the CheckMate 009/010/025 clinical trials consented to an institutional review board (IRB) approved protocol to participate in the respective clinical trials and to have their samples collected for tumor and germline sequencing.

The two patient cohorts from DFCI are defined in Supplementary Data 1. Cohort 1 was used to define the regulatory landscape across the different RCC histologies. On the other hand, cohort 2, was used primarily for the Allelic Imbalance analysis to increase the power of our

statistical analysis. In cohort 1, 28 of 30 (93%) tumor samples were obtained at radical nephrectomy, and 2 of 30 (7%) were from metastatic sites (Supplementary Data 1). Cohort 2 was used for the allelic imbalance, which consisted of 14 patients from cohort 1 in addition to 12 additional independent patients with ccRCC. Overall, 42 total samples were profiled (Supplementary Fig. S1). All specimens were reviewed by pathologists with expertise in genitourinary malignancies (TD, AF, SM, AA, MH, SS) to confirm the diagnosis, histological subtype, tumor grade, and stage (Supplementary Data 1). An additional cohort of 43 samples derived from 23 patients with ccRCC was characterized to study the effect of EPAS1 binding at the GWAS risk variant rs4903064 using *EPAS1* ChIP-Seq (Supplementary Data 1).

### Chromatin immunoprecipitation for histone and TF marks

We performed chromatin immunoprecipitation (ChIP) for histone marks (H3K27ac and H3K4me2) in primary human tumors[89]. Briefly, 20–30 mg of frozen tissue was pulverized using the CryoPREP dry impactor system (Covaris). The tissue was then fixed using 1% formaldehyde (Thermo Fisher) in PBS for 10 min at room temperature and then quenched with 125 mM glycine. The tissue was then lysed in ice-cold lysis buffer (50 mM Tris, 10 mM, EDTA, 1% SDS with protease inhibitor), and the chromatin was sheared to 300–800 base pair using the Covaris E220 sonicator (105-watt peak incident power, 5% duty cycle, 200 cycles/burst). Five volumes of dilution buffer (1% Triton X-100, 2 mM EDTA, 150 mM NaCl, 20 mM Tris-HCl pH 8.1) were then added, and a portion was taken for DNA preparation. 5–20 µg DNA equivalent was then incubated with antibodies (H3K27ac, Diagenode, C15410196, 1 µg/IP dilution; H3K4me2, Diagenode, C15410035, 1 µg/IP dilution), along with protein A and protein G beads (LifeTechnologies) with constant mild shaking at 4 °C overnight. The beads were then washed three times each with Low-Salt Wash Buffer (0.1% SDS, 1% Triton X-100, 2 mM EDTA, 20 mM Tris-HCl pH 7.5, 150 mM NaCl), High-Salt Wash Buffer (0.1% SDS, 1% Triton X-100, 2 mM EDTA, 20 mM Tris-HCl pH 7.5, 500 mM NaCl), and LiCl Wash Buffer (10 mM Tris pH 7.5, 250 mM LiCl, 1% NP-40, 1% Na-Doc, 1 mM EDTA) and rinsed with TE buffer (pH 8.0) once. For EPAS1 (HIF2-α) ChIP (Abcam, ab199, 1:100 dilution), the same amount of tissue was additionally fixed with 0.2 mM DSG (Thermo Fisher, Catalogue number: 20593) for 25 min followed by 1% fixation with FA for 20 min. The rest of the protocol per above.

Sequencing libraries were generated from purified input and IP sample DNA using the ThruPLEX-FD Prep Kit (Rubicon Genomics). Libraries were sequenced using 150-base paired-end reads on an Illumina platform (Novogene). Data quality is shown in Supplementary Data 26.

### ATAC-seq

Briefly, 20 mg of frozen tissue were resuspended and dounced in 1000 µl of HB. Nuclei were filtered using a 70-µm Flowmi strainer, isolated using iodixanol density-gradient centrifugation method, and washed with RSB buffer[61]. In all, 50,000 nuclei were resuspended in 50 µl of transposition mix (2.5 µl transposase (100 nM final), 16.5 µl PBS, 0.5 µl 1% digitonin, 0.5 µl 10% Tween-20, and 5 µl water) by pipetting up and down six times. Transposition reactions were incubated at 37 °C for 30 min in a thermomixer with shaking at 1000 r.p.m. Reactions were cleaned with Qiagen columns. Libraries were amplified were prepared using the Omni-ATAC protocol[90,91] and sequenced on an Illumina platform (Novogene) using 150-base paired-end reads. Data quality is shown in Supplementary Data 26.

### ChIP-seq data analysis

ChIP-sequencing reads were demultiplexed using Illumina bcl2fastq v2.18 and aligned to the human genome build hg19 using the Burrows-Wheeler Aligner (BWA) version 0.7.15[92]. Non-uniquely mapping and redundant reads were discarded. MACS v2.1.1.20140616[93] was used for

ChIP-seq peak calling with a *q*-value (FDR) threshold of 0.01. ChIP-seq data quality was evaluated by a variety of measures, including total peak number, FrIP (fraction of reads in peak) score, number of high-confidence peaks (enriched >tenfold over background), and percent of peak overlap with DHS peaks derived from the ENCODE project. ChIP-seq peaks were assessed for overlap with gene features and CpG islands using annotatr[94]. IGV[95] was used to visualize normalized ChIP-seq read counts at specific genomic loci. ChIP-seq heatmaps were generated with deepTools[96] and show normalized read counts at the peak center ± 2 kb unless otherwise noted. Overlap of ChIP-seq peaks was assessed using BEDTools. Peaks were considered overlapping if they shared one or more nucleotides.

### Annotation of histology-specific enriched ChIP-seq peaks
Sample-sample clustering, principal component analysis, and identification of lineage-enriched peaks were performed using Mapmaker (https://bitbucket.org/cfce/mapmaker), a ChIP-seq analysis pipeline implemented with Snakemake[97]. ChIP-seq data from ccRCC, chRCC, and pRCC were compared to identify H3K27ac and H3K4me2 peaks with significant enrichment in either histology. A union set of peaks for each histone modification was created using BEDTools v2.26.0. narrowPeak calls from MACS were used for H3K27ac and H3K4me2. The number of unique aligned reads overlapping each peak in each sample was calculated from BAM files using BEDTools. Read counts for each peak were normalized to the total number of mapped reads for each sample. Quantile normalization was applied to this matrix of normalized read counts. Using DESeq2[98], lineage-enriched peaks were identified at the indicated FDR-adjusted *P* value ($P_{adj}$) and log2 fold-change cutoffs (H3K27ac and H3K4me2, *P*-adjusted <0.001, log2 fold change >3). Unsupervised hierarchical clustering was performed based on the Spearman correlation between samples. Principal component analysis was performed using the prcomp R function. Enriched de novo motifs in differential peaks were detected using HOMER version 4.7. The top non-redundant motifs were ranked by $P_{adj}$. The GREAT tool[33] was used to assess for enrichment of Gene Ontology (GO) and MSigDB perturbation annotations among genes near differential ChIP-seq peaks, assigning each peak to the nearest gene within 500 kb.

### RNA methods
RNA was extracted using the Qiagen RNeasy Mini Kit (Cat No./ID: 74104) as recommended, from frozen tumor samples adjacent to those samples used for ChIP. RNA-seq libraries were constructed from 1 µg total RNA using the Illumina TruSeq Stranded mRNA LT Sample Prep Kit according to the manufacturer's protocol. Barcoded libraries were pooled and sequenced on the Illumina HiSeq 2500 generating 50-bp paired-end reads. FASTQ files were processed using the VIPER workflow[99]. Read alignment to human genome build hg19 was performed with STAR[100]. Cufflinks was used to assemble transcript-level expression data from filtered alignments[101].

### DNA extraction and mutation analysis
DNA was either (1) extracted from OCT-frozen primary tissue using the Qiagen QIAamp DNA Mini Kit (Cat No./ID: 51304) (*n* = 9); or (2) extracted from tumor regions consisting of at least 20% tumor cells from unstained slides using the QIAamp DNA FFPE Tissue Kit (Qiagen) according to the manufacturer's instructions (*n* = 19). DNA quantification was performed by Nanodrop and Pico-Green assays.

Mutational analysis was performed using our institutional CLIA-certified targeted panel sequencing, Oncopanel[102] without normal DNA analysis. Called variants were excluded if observed at a frequency ≥0.1% in the Exome Aggregation Consortium (ExAC) database[103], as they were considered likely germline variants. Loss-of-function variants were defined as nonsense mutations, frameshift insertions or deletions, or splice site alterations affecting consensus nucleotides (Supplementary Data 3). Missense mutations were only included if deemed pathogenic by (1) both SIFT[104] and Polyphen-2[105] and (2) reported as oncogenic or likely oncogenic in cBioPortal (TCGA Pan-cancer Atlas studies)[106,107]. CNVs were called by Oncopanel, and here we retained only homozygous deletions (Supplementary Data 4).

### Histology-specific expression of TFs
Normalized gene expression was computed for all 1631 known TFs[108,109] in all 28 RCC samples with RNA-Seq data (11 chRCC, 11 ccRCC, and 6 pRCC). To identify TFs whose expression was significantly higher in one histology than another, we compared the normalized TF expression using the Wilcoxon rank-sum two-sample one-sided test. TFs with an average normalized expression less than 10 FPKM in the histotype with the highest expression were excluded. Multiple hypothesis correction was done using Benjamini–Hochberg[102]. Unsupervised clustering of the samples using the 105 differentially expressed TFs, that had log2 fold change ≥1, FDR < 0.1, and minimum of 10 transcripts per million was performed to separate the RCC samples according to histology (Supplementary Fig. S5A and Supplementary Data 5 and 6).

### Cancer core transcription-factor specificity algorithm
Cancer Core Transcription factor Specificity (CaCTS) method is developed and described in detail in ref. [110], which compares the expression level of a TF in each RCC histotype with all other cancer types. The CaCTS algorithm has been previously developed as an R package and has been deposited in GitHub (https://github.com/lawrenson-lab/CaCTS) and Zenodo (https://doi.org/10.5281/zenodo.5234007).

Briefly, we used pan-cancer RNA-sequencing data from The Cancer Genome Atlas to identify RCC histology-specific candidate master TFs. Considering a set of transcription factors detailed previously[108,109], we got a list of 1671 unique TFs. We selected only the ones expressed in the TCGA pan-cancer dataset[49]. CaCTS score defines cell-specific TFs for each cancer type based on the RNA expression information. The specificity of expression of each TF, or "CaCTS score", was calculated by comparing its expression level in the query tumor type to that in the remaining TCGA tumor types. A high CaCTS score is therefore assigned to factors with high-level expression in the query tumor types as compared to background dataset. The output of the CaCTS algorithm is a list of all TFs ranked by CaCTS scores in each of the three RCC subtypes. The main procedure consists in a Jensen-Shannon Divergence (JSD) score. We first adjusted the normalized expression values, to handle negative values, by shifting the values to 0 and the maximum expression value. Then, to define a representative sample to each tumor type, we applied the mean of the expression values considering each tumor type individually. Mean values of 0 were replaced with $1 \times 10^{-17}$. We followed the same principles described by D'Alessio et al.[13] aiming to quantify the divergence between the transcription-factor expression across different cell types. We created two same-sized vectors to represent the observed pattern and the "ideal" pattern. For the observed pattern, the vector was formed by values from the expression mean profiles of the query cell type and the background dataset. Each element in this vector were divided by the sum of all elements. For the "ideal" pattern, the vector was formed by a value of 1 at the position equivalent to that of the query cell type and zeroes at all other positions. For example, considering ccRCC as query and the other 33 cancer types as background, the position 1, that corresponds to ccRCC position, was formed by a value 1 and the other positions were formed by value 0. From these two vectors, the JSD function was performed with the R package jsd version 0.1 and a cell-type-specificity score was obtained for each TF. The candidate list for each cancer type was defined by considering the top 5% of high JSD and top 5% of ranked expression mean. This analysis identified 36 TFs that were highly and uniquely expressed in one of the three RCC histotypes in comparison

to TCGA cancers (Fig. 3B, Supplementary Fig. S5B–S5D, and Supplementary Data 7).

## Superenhancer rank analysis

Super-enhancers (SEs) and typical enhancers (TEs) were identified using ROSE2 (https://pypi.org/project/rose2/)[17] (Supplementary Fig. S9 and Supplementary Data 8). Both SEs and TEs were ranked based on integrated ChIP-seq signal[17], where the SE with rank 1 has the highest ChIP-seq signal. SEs were then associated to the nearest gene, giving an SE rank score for each gene[17]. For each transcription-factor-sample pair, we identified the minimum TF SE rank for all SEs associated with that TF, or assigned the maximum SE rank found in all samples (1477) if the TF was not associated to any SE in that sample. TFs with an average superenhancer rank greater than 1000 were not considered further. We then compared the SE ranks for each TF and histology against the other histologies using the Wilcoxon rank-sum two-sample one-sided test. Multiple hypothesis correction was done using the Benjamini–Hochberg Procedure, and $FDR \leq 0.10$ was considered significant. In all, 4073 SE region-associated genes were detected in at least one RCC sample, including 377 TFs. Unsupervised hierarchical clustering of the SE-associated TFs partitioned the samples according to their histology (Supplementary Fig. S5E) as did similar analysis of the 119 TFs with differential SE rank by histology (FDR < 0.1) (Supplementary Fig. S5F and Supplementary Data 9).

## ATAC-seq normalization

Publicly available ATAC-seq data from 16 ccRCCs and 34 pRCCs[61], and newly generated ATAC-seq data for seven chRCC samples were integrated and normalized. chRCC ATAC-seq peaks sets were normalized[111] by computing a "score per million" (spm), which is the individual peak score ($-\log10(P$ value)) by the total sum of all peak scores in the sample multiplied by a million, and peaks with spm ≤5 were filtered out. Second, to generate the chRCC-specific ATAC-seq peak set, we conserved regions covered by at least two samples, breaking up peaks if necessary. Lastly, we removed regions that overlap repeat regions considering the UCSC Table browser (http://genome.ucsc.edu/cgi-bin/hgTables) "Repeats" group and "Repeatmasker" track. Only regions that contain the full length as $N$ were removed.

## Clique enrichment score analysis

SE-associated TFs are typically wired in auto-regulatory networks, called "cliques", that cooperatively regulate their own expression and expression of an extended network of genes to create the transcriptional fingerprint of the tissue[65]. To identify highly interconnected TFs, we integrated H3K27ac and ATAC-seq data and used Coltron to identify cliques and determine the clique enrichment score (CES). CES for each TF was calculated using clique assignments performed using Coltron[112]. Coltron uses a motif-based approach employing both SE data and normalized ATAC-seq data to restrict the motif search to regions of open chromatin, to build networks of transcription factors. It identifies cliques of various sizes reflecting putative interaction between TFs in driving TF expression. The CES equals the fraction of cliques containing each expressed TF. Using the CES, we performed clustering analysis with the following parameters: distance = Canberra, agglomeration method = ward.D2. To select RCC histology-specific TFs using the CES, we restricted to a minimum average CES of 0.05, i.e., a TF has to be present, on average, in at least 5% of all cliques in a given RCC histological subtype. 167 TFs had a positive CES in one or more tumors, and unsupervised clustering of the samples based on TF CES separated samples by histology (Supplementary Fig. S5G). In total, 87 of the 167 TFs (52%) showed differential connectivity between the three histologies, meaning that on average, they participated in a significantly larger number of cliques in one histology compared to the other (one-sided Mann–Whitney $U$ test of one histotype vs. the rest,

FDR < 0.1, minimum CES = 0.05) (Supplementary Fig. S5H and Supplementary Data 10–11).

## Master TF analysis

These four analyses (Differential expression, CaCTS, SE rank, and CES) identified more than 200 candidate TFs showing a histology-specific association in one or more analyses. However, each analysis has potential limitations. Differential gene expression is not discriminative enough and is expected to have too many false positives. SE rank analysis relies heavily on enhancer-gene assignment, which is biased against long-distance interactions. CES analysis is a motif-based approach and depends on the prior knowledge of TF motifs, which are incompletely characterized. Finally, CaCTS is designed to compare pan-cancer TF expression, so it can miss TFs that are also important in other cancer types, which is why we used a meta-analysis approach where the TF has to be significant in one of the three analyses (CaCTS, SE rank, and CES) as well as significantly differentially expressed between the three histologies (Supplementary Data 12).

## Immunohistochemistry

For immunohistochemistry (IHC), 5-mm FFPE tissue sections were deparaffinized, rehydrated, and subjected to heat-induced antigen retrieval (0.01 mol/L sodium citrate tribasic dihydrate, pH 6.0 or EDTA retrieval buffer, pH 9.0) followed by treatment with 3% $H_2O_2$ for 15 min at room temperature to block endogenous peroxidase activity. Sections were incubated overnight at 4 °C with primary antibodies against mouse HNF1β (Santa Cruz, sc-130407, 1:50 dilution), mouse BHLHE41 (Thermofisher, TA806146, 1:200 dilution), rabbit NKX6.1 (Cell signaling, #54551, 1:50 dilution), rabbit ZNF395 (Lsbio, LS-B5647-100, 1:500 dilution). Sections were then stained with appropriate secondary antibodies per manufacturer recommendations (ImmPRESS HRP anti-rabbit and anti-mouse IgG Polymer detection kits: MP-7451; MP-7402, Vector laboratories). Slides were counterstained with hematoxylin, dehydrated, and mounted. Slides were scanned at ×20 magnification and positively stained tumor cells for HNF1β, BHLHE41, NKX6.1, or ZNF395 were determined.

## Cell lines

786-O cell line was obtained from a certified commercial vendor (ATCC, CRL-1932) and grown in DMEM (Invitrogen) supplemented with 10% fetal bovine serum (Sigma), 100 U/mL penicillin, and 100 µg/mL streptomycin (Invitrogen). No further authentication was performed. Cell lines tested negative for mycoplasma contamination.

## FOXI1 overexpression in 786-O cell line (FOXI1 OE)

The full-length *FOXI1* open reading frame (ORF) without stop codon was amplified by gene-specific cloning primers (Supplementary Data 27) using cDNA from an RCC cell line which expressed FOXI1 endogenously (Obtained from L. H.'s lab) and cloned into mammalian expression lentiviral vector (pFUGW) derivative (Addgene #52962) AgeI and BamHI sites. Correct *FOXI1* ORF sequence was verified by Sanger sequencing, expression level was measured by qRT-PCR and correct protein size was verified by immunoblot technique. Stable FOXI1 overexpressing cell lines were created by lentiviral transduction using the FOXI1 containing transfer vector and pMD2.G (Addgene #12259) and psPAX (Addgene#12259) lentiviral packaging vectors. Transduced cells were selected and maintained under blasticidin containing (10 µg/ml) selective culture media.

## EPAS1 shRNA knockdown in 786-O cell line (EPAS1 KD)

Short hairpin RNAs (shRNA) were designed against the FOXI1 3′ UTR region using the Broad Gene Perturbation Web Portal (GPP): https://portals.broadinstitute.org/gpp/public/ and non-human targeting control (NTC) gRNA was included and cloned (detailed protocol available at http://www.broadinstitute.org/rnai/public/resources/protocols).

Briefly, for each shRNA (Supplementary Data 27) complementary single-stranded oligonucleotides were synthesized (Invitrogen) with cloning sites, phosphorylated, annealed, and shRNA cassettes were ligated into pLKO.1 (Addgene #8453) shRNA expression vector containing puromycin selection marker. After bacterial transformation (Stbl3, Invitrogen) individual clones were picked and regenerated, and correct shRNA sequences were verified by Sanger sequencing. Suppression effects were tested by qRT-PCR and immunoblot techniques.

Lentiviral particles were generated for each shRNA experiment by transforming 786-O cells with shRNA transfer vectors and lentiviral packaging mix (pMD2.G psPAX). Lentiviral particle containing media were collected and filtered using 0.45-µm pore size syringe filters (Corning) after 48 h post transfection and used for treatment of previously plated 786-O cell line. Media was changed after 24 h and replaced by puromycin (2 µg/ml) containing selective culture medium. Puromycin-resistant cells were maintained and collected after 3 days of selection and total RNA was isolated using RNeasy Mini kit (Qiagen).

### Creation of double manipulated cell lines (FOXI1 OE/EPAS1 KD)

786-O/FOXI1 OE cells were cultured and maintained in DMEM media as described above. Cells from the two different conditions were seeded in parallel six-well plates and 24 h later, cells were infected with lentivirus containing shRNA targeting EPAS1 and negative control shRNA. Cells were selected by puromycin for 72 h following infection, equal cell numbers were harvested for RNA extraction and immunoblotting.

### Gene expression analyses

For targeted gene expression, qRT-PCR 500 ng total RNA (Qiagen) was reverse transcribed (High Capacity Reverse transcription kit, Life-Technologies) and cDNA was diluted (20×). SYBR Green assay was performed on Light Cycler 480 instrument (2x Probe Master Mix, Roche). All primer sequences are listed in Supplementary Data 27. Relative gene expression was calculated based on the ddCT method (Pfaffl 2001). Each sample was measured by two technical replicates. *ACTB* gene was used as housekeeping genes to normalize the samples.

For gene expression profile analysis, RNA-Seq was performed from the different conditions using poly A capture method from total RNA by Novogene sequencing service.

### Chromatin and gene expression allelic imbalance analysis

DNA samples from three pRCC and ten ccRCC samples were genotyped on a commercially available genotyping array (Infinium Global Screening Array-24, version 1.0; Illumina) at the Broad Institute Genomic Services, Cambridge, Massachusetts. We used a total of 642,824 SNP markers. H3K27ac ChIP-seq reads from the same 10 ccRCC and 3 pRCCs were analyzed for the imbalance of heterozygous SNP alleles using stratAS (https://github.com/gusevlab/stratAS; Gusev et al., submitted; https://www.biorxiv.org/content/10.1101/631150v1.full.pdf, deposited in Zenodo (https://doi.org/10.5281/zenodo.7373647)), hereby denoted chromatin allelic imbalance. Several upstream steps were performed to boost power and accuracy of chromatin allelic imbalance detection.

First, genotypes were imputed from SNP array data with STITCH[113], using the 1000 Genomes Phase 3 reference panel. Subsequent analyses were limited to SNPs with allele fraction >5% in the Haplotype Reference Consortium version 1.1[102]. Genotypes were then phased with Eagle2 using the Sanger Imputation Service (https://imputation.sanger.ac.uk/). Heterozygous SNPs were filtered for mapping bias using the WASP pipeline[114] and allele-specific read counts were tabulated using ASEReadCounter from the Genome Analysis Toolkit v3.8[115].

Briefly, stratAS identifies chromatin allelic imbalance—and differential allelic imbalance between two-sample groups—by modeling the reads from heterozygous SNPs with a beta-binomial distribution (https://doi.org/10.1101/631150v1.full.pdf). stratAS takes advantage of haplotype phasing to sum read counts from heterozygous SNPs from each phased haplotype for each individual within a given ChIP-seq peak. stratAS models the reads from individual $i$ overlapping heterozygous germline SNP $j$ as: $R_{alt,i} \mid R_{ref,i} BetaBin(\pi_i, \rho_{ij})$, where $\pi$ is the mean allelic ratio and $\rho$ is a locally-defined, per-individual sequence read correlation parameter reflecting overdispersion.

We tested differential chromatin imbalance between ccRCC and pRCC RCC by the likelihood ratio test between the models $\pi_{clearcell,j} = \pi_{pRCC,j}$ and $\pi_{clearcell,j} \neq \pi_{pRCC,j}$, maximizing the likelihood of each model by a standard 1-dimensional optimization.

Copy number profiles were estimated from read representation in ChIP-seq inputs using qdnaseq v1.18. Copy number profiles were used in the modeling of the overdispersion parameter $\rho$. This is motivated by larger degrees of overdispersion in regions of cancer-associated copy number alterations. $\rho$ is estimated for each individual from all heterozygous read-carrying SNPs across ten declines of estimated copy number levels stratAS params.R script, with the following options:−min_snps 50, min_cov 5,−group_snp TRUE,−group 10.

We tested variants within a consensus set of H3K27ac peaks. This peak set was derived by dividing the genome into 50 bp windows and including any window with peaks in two or more samples. We restricted tests to peaks with at least one read supporting each haplotype (−min_cov 1). The following additional parameters were set for the stratas.R script:−max_rho 0.2,−window −1, min_cov 1, and−fill_cnv TRUE. $P$ values for allelic imbalance, and differential imbalance between ccRCC and pRCC, were corrected for multiple hypothesis testing as follows. First, all SNPs within a given peak were adjusted by Bonferroni correction. The corrected $P$ value of the most significant SNP was taken as the $P$ value for imbalance of the peak. We then obtained FDR-adjusted q-values for all peaks using the q-value R package (v4.0.1). Peaks with $q < 0.05$ were considered significantly imbalanced. Differentially imbalanced peaks were grouped into categories for motif enrichment analysis. Peaks with ccRCC-specific chromatin imbalance were defined as those with (1) a significant imbalance in ccRCC, (2), significant differential imbalance between ccRCC and pRCC, and (3), no significant imbalance in the stratAS analysis of pRCC alone. Conversely, a set of peaks with pRCC-specific imbalance was defined by (1) chromatin imbalance in pRCC, (2) differential chromatin imbalance between the two groups, and (3), no significant chromatin imbalance in ccRCC. These two groups were analyzed for motif enrichment using homer v4.7. The background for enrichment was specified as all peaks imbalanced in pRCC and in ccRCC in the separate stratAS analyses of these groups. The top three "known results" motif categories were reported.

The union set of chromatin-imbalanced peaks (i.e., in pRCC, ccRCC, or in the combined or differential comparisons) were assessed for enrichment of significant SNPs from genome-wide association studies (GWAS) of RCC[58], prostate cancer (6), psoriasis[116], cardiovascular disease, diagnosis of hypertension[117], self-reported hypercholesterolemia, type 2 diabetes mellitus[118], self-reported hypothyroidism[119], all autoimmune diseases, dermatologic disorders, diagnosis of asthma, respiratory disease, and diagnosis of allergy or eczema. For each GWAS, the number of genome-wide significant SNPs overlapping chromatin-imbalanced H3K27ac peaks was counted and divided by the number of base pairs covered by these peaks. The same process was performed with a background set of peaks randomly sampled over 10,000 iterations from all RCC H3K27ac peaks. The relative overlap of chromatin-imbalanced peaks was divided by the relative overlap of background H3K27ac peaks to estimate relative enrichment. Enrichment was also calculated relative to random peaks, matched to the chromatin-imbalanced peaks for number, peak size, and chromosome. Empiric one-sided $P$ values were calculated from the iterations.

## Reporting summary

Further information on research design is available in the Nature Portfolio Reporting Summary linked to this article.

## Data availability

Sequencing reads are aligned to the human genome build hg19. The DFCI datasets generated in this study, including ChIP-seq, RNA-seq, and ATAC-seq data (R0929TAT, R0957TAT, R0998TAT, R1097TAT, R1149TAT, R1152TAT, R991T1AT) have been deposited in the Gene Expression Omnibus (GEO) database under accession code GSE188486. The TCGA Kidney Renal Clear Cell Carcinoma (KIRC) publicly available data used in this study are available in a public repository from the Broad Institute Firehose Pipeline (http://gdac.broadinstitute.org). All clinical and correlative sequencing data from publicly available. CheckMate 009/010/025 publicly available data used in this study are available in the European Genome-Phenome Archive database under accession codes EGAS00001004291 and EGAS00001004292[120]. ATAC-Seq data publicly available data used in this study are available in the Genomic Data Commons database (https://gdc.cancer.gov/about-data/publications/ATACseq-AWG)[61]. The remaining data are available within the Article or Supplementary Information. Any other queries about the data used in this study should be directed to the corresponding authors of this study. Source data are provided with this paper.

## Code availability

Algorithms used for data analysis are all publicly available from the indicated references in the Methods section.

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

## Acknowledgements

We are grateful for all our patients treated at Dana-Farber Cancer Institute. This work is supported by the United States Department of Defense (Idea Award, W81XWH-19-1-0554— M.L. Freedman, M.M. Pomerantz, A. Nassar, S. Abou Alaiwi). M.L. Freedman is supported by the Claudia Adams Barr Program for Innovative Cancer Research, the H.L. Snyder Medical Research Foundation, the Cutler Family Fund for Prevention and Early Detection, the Donahue Family Fund, the Department of Defense Awards W81XWH-21-1-0234 (M.L.F.), W81XWH-21-1-0339 (M.L.F.), W81XWH-19-1-0554 (M.L.F.), NIH Awards R01CA251555 (M.L.F.), R01CA227237 M.L.F.), and a PCF Challenge Award. T.K. Choueiri is supported in part by the Dana-Farber/Harvard Cancer Center Kidney SPORE (2P50CA101942-16) and Program 5P30CA006516-56 (T.K.C.), the Kohlberg Chair at Harvard Medical School and the Trust Family, Michael Brigham, Pan Mass Challenge, Hinda and Arthur Marcus Fund and Loker Pinard Funds for Kidney Cancer Research at DFCI. M.M. Pomerantz is also supported by John and Ann Hall; Rebecca and Nathan Milikowsky Family Foundation.

## Author contributions

A.H.N., S.A.A., T.K.C., D.J.K., and M.L.F. conceived the study; A.H.N., S.A.A., S.C.B., E.A., R.I.C., and M.AS.F. analyzed the data under the joint supervision of D.J.K., M.L.F., and K.L.; A.H.N and S.A.A. wrote the manuscript under the under the joint supervision of T.K.C., D.J.K., and M.L.F.; A.H.N. and S.A.A. performed ChIP-seq, ATAC-seq, RNA-seq, and SNP array experiments with assistance from T.E.Z. and J.S.; S.Spisak., V.T. performed shRNA experiments under the supervision of M.L.F. S.G. assisted with the genotype imputation; A.F., A.A., T.D., S.G.M., A.A.,

S. Signoretti., and M.S.H. reviewed the pathology slides; S.C.B., E.A., and A.G. analyzed the allelic imbalance data. D.A.B., H.D., M.H., S.S., Y.H., J.H., G.B., J.E.B., E.O., C.B., P.V.N., G.M.L., M.P., E.H., and M.A. assisted with procurement of clinical samples. All authors assisted in writing and revising the manuscript. T.K.C., D.J.K., and M.L.F. are co-senior authors for this work.

## Competing interests

The authors declare no competing interests.

## Additional information

[1]Department of Hematology/Oncology, Yale New Haven Hospital, New Haven, CT 06510, USA. [2]Department of Medicine, Brigham and Women's Hospital, Boston, MA 02115, USA. [3]Department of Medical Oncology, Dana-Farber Cancer Institute, Boston, MA 02215, USA. [4]Center for Functional Cancer Epigenetics, Dana-Farber Cancer Institute, Boston, MA 02215, USA. [5]Women's Cancer Research Program at the Samuel Oschin Comprehensive Cancer Institute, Cedars-Sinai Medical Center, Los Angeles, CA, USA. [6]Division of Gynecologic Oncology, Department of Obstetrics and Gynecology, Cedars-Sinai Medical Center, Los Angeles, CA, USA. [7]Center for Bioinformatics and Functional Genomics, Samuel Oschin Comprehensive Cancer Institute, Cedars-Sinai Medical Center, Los Angeles, CA, USA. [8]The Eli and Edythe L. Broad Institute, Cambridge, MA 02142, USA. [9]Department of Pathology, Brigham and Women's Hospital, Boston, MA 02115, USA. [10]Translational Immunogenomics Lab, Dana-Farber Cancer Institute, Boston, MA, USA. [11]McGraw/Patterson Center for Population Sciences, Dana-Farber Cancer Institute, Boston, MA 02115, USA. [12]These authors contributed equally: Amin H. Nassar, Sarah Abou Alaiwi, Sylvan C. Baca. ✉e-mail: toni_choueiri@dfci.harvard.edu; dk@rics.bwh.harvard.edu; mfreedman@partners.org

