## [Peer Review File · Nature Communications]

Epigenomic Charting and Functional Annotation of Risk Loci in Renal Cell CarcinomaReviewers' Comments:

Reviewer #1:

Remarks to the Author:

This manuscript has collected a rich datasets characterizing epigenomic landscapes of clear cell, papillary, and chromophobe RCC cancer cells by using ChIP-seq, ATAC-Seq, RNA-seq, and SNP arrays. They have shown the subtype-specific enhancer regulation profile, predicted master TFs and validated a germline SNP that recognizes HIF-2 α with different binding affinity. Although the comprehensive epigenomic atlas in primary patient samples undoubtedly provide a valuable resource for the research community, it is descriptive to me. There are significantly incremental concerns about the uncertain quality control, data interpretation and lack of functional validation, which dampened my enthusiasm to recommend for publication at the current status. I hope that the authors will find the enclosed comments helpful.

Major comments

1. There are no normal tissues or non-tumor tissues collected accompanying with tumor harvest. Therefore, when using the term of "tumor-specific", actually there are many H3K27ac marked enhancers that are conserved in normal developmental tissues. Those "subtype-specific" epigenetic signatures, GSEA and pathway analysis should be re-justified by comparing with normal kidney tissue data. I am sure here are plenty datasets available as normal tissue controls at GEO, GTEX or other resources. This is also related to the justification of using one single cell line 786-O for validating the SNP. The authors claimed that this is a typical ccRCC cell line. Did they characterized the epigenomic landscape of this cell line to cluster it with ccRCC with more correlation? I understand the difficulty to translate primary patients samples to cell line, but It is important to choose the right cell type sharing the same epigenomic signature, which is the key point of the this manuscript.
2. Although multi-Omics data were collected, insufficient integrative analysis as actually conducted. I strongly recommend more validation analysis by using different datasets. For instance, motif enrichment analysis using H3K27ac marked peaks is not powerful strategy given most of the histone modification markers bound broad peaks, great diluting the statistic powder of motif enrichment. However, ChIP-seq of TFs and ATAC-seq peaks were sharp and also correlated with H3K27ac. Since several TFs were predicted, the authors should carry out ATAC-seq motif enrichment analysis as an independent approach to validate the predicted motifs from H3K27ac analysis. Additionally, ATAC-seq with high sequencing resolution can provide footprint signature to support the direct binding. Similarly, CRISPR dropout screens have been extensively used to identify dependency genes and master TFs in kidney cancer. Depmap data should be explored to further support the functional value of the "master TFs". These integrative analysis should further strengthen the data mining and better support the main conclusion.
3. The authors claimed to predict 50 top master TFs which out any other data support how they are controlling the subtype-identify and transcription. The binding occupancy switch at least should be tested in cell lines from different subtypes.
4. Figure 4 starts with ambition but ends of lack of support from GWAS study and functional validation again. The authors should incorporate the previous GWAS hits from large cohort studies with the functional noncoding loci marked by H3K27ac and allele-specific expression (ASE) to further extend the power of epigenomic landscape. Whether the imbalanced binding of TFs lead to ASE of target genes from RNA-seq?
5. The entire validation of a single SNP at DPF3 locus relies on binding difference. The most direct evidence will be conducting genome editing of either risk or non-risk allele in a relevant cell lines and examine the impact on transcription.
6. Quality control (QC) of each datasets should be provided in supplementary figures to show the reproducibility and stats.
7. Figure fonts and character styles are all different. Some of the fonts are too small for me to visualize.

Reviewer #2:

Remarks to the Author:

In this paper, Nassar et al. reported an interesting data set of H3K27ac ChIP-seq, H3K4me2 ChIP-seq, ATAC-Seq, RNA-seq, and SNP arrays on human renal cell carcinoma. Through integrated analysis, they identified 50 candidate histology-specific master transcription factors in three main RCC subtypes (ccRCC, pRCC and chRCC). They further evaluated and validated two transcription factors in 786-O cells. By assessing ChIP-seq data, they also annotated risk SNPs identified by a GWAS for RCC. This manuscript contains an abundance of epigenomic profiling data from patient samples and represents an important resource for the field. The figures are generally well prepared. However, I have several concerns about this current version of the paper.

Major concerns:

1. In contrast to ccRCC and chRCC, a relatively few pRCC samples were performed for ChIP-seq and ATAC-Seq in this study. But pRCC have two main histologic types, type 1 and type 2, which were shown to be different types of renal cancer characterized by specific genetic alterations and biologic behavior. The specific types of pRCC in cohort 1 should be described in detail. If these pRCC all or mainly belong to one pRCC type, it is hard to identify pRCC-specific master transcription factors based on these samples in this study. And more samples of the other type should be added. Also, the analysis about pRCC should be divided into type 1 and type 2.

2. Through integrated analysis, the author identified 50 candidate histology-specific master TFs in three main RCC subtypes, but the expression of these TFs in patient tissue samples need to be validated through IHC or immunofluorescence staining, at least for selected TFs. Furthermore, the potential regulatory circuitries in driving different RCC histologic subtypes should be explored. For example, (1) the binding profiles of these TFs in each RCC subtype; (2) co-regulated relationship; (3) TF hotspots; (4) co-regulated SE domains and the key regulators in the regulatory network.

3. Are the histology-specific master TFs identified in this study associated with patient clinical features and prognosis?

4. The validation of master TFs should be performed on patient derived organoids or at least primary patient cancer cells rather than 786-O. Furthermore, for the most important finding such as FOXI1 and EPAS1, in vivo validations should be considered.

5. After overexpression or suppression of the master TFs, only gene expression changes were evaluated. Further functional experiments such as angiogenesis, cell proliferation and migration assays were also needed. Moreover, the changes of biomarkers for ccRCC (eg, CAIX) and chRCC (eg, CK7, CD117) after the manipulation of TF expression should also be evaluated.

6. The related results of FOXI1 OE and EPAS1 KD in cells should be added in supplementary figures and tables.

7. VHL-wild type cells or models should be added in the validation of master TFs for ccRCC.

8. Master TFs for pRCC (eg, HNF1B) also need to be validated, rather than discussed in the discussion section.

Minor concerns:

1. Fig 2D, HIF-1a should be changed to HIF-2a.

2. Page 9, line 200-201, "FOXI1 scored as a chromophobe-specific TF in 3/4 master TF analyses (Figure 3B)". Figure 3B fails to reflect this result.

3. Page 9, line 204-205, "We also manipulated the expression of EPAS1 as a second target as it was highly specific for ccRCC in our integrative analysis.". This result is shown by Figure S7B.
4. Page 10, line 213-214, "Of note, the FOXI1 OE/EPAS1 KD cell line did not have significant expression differences compared to FOXI1 OE only.". Perhaps I am missing it, I could not find this result. Also, the expression differences between FOXI1 OE/EPAS1 KD and EPAS1 KD cell lines should be added in supplementary figures and tables.
5. Figure S7, the expression changes of FOXI1 and EPAS1 protein should be added.
6. Figure S8, Volcano plot of RNA-Seq data showing differentially expressed for 786-O CTRL vs. FOXI1 OE should be added.
7. Figure S8, figure S8C was missed.
8. Figure S8D, the comparative objects should be added.
9. Page 12, line 273, HIF2-alpha should be changed to HIF2- α .

Reviewer #3:

Remarks to the Author:

This is a well-executed comprehensive epigenetic study that has identified 50 candidate histology-specific master transcription factors (TF) among renal cancer carcinoma (RCC). The work is significant to fill the knowledge gap in the area. The study is unique to distinct three histology types of RCC, i.e., clear cell (ccRCC), papillary (pRCC), and chromophobe (chRCC) from 42 patient tumor sample analysis.

Figure 3: the authors cross validated that overexpression of a single chRCC master transcription factor, FOXI1, in ccRCC cell line 786-O, with or without knockdown of EPAS1, led to marked expression changes driving the cell line to be more like a chRCC cell line without any modification to the set of mutations present in 786-O cells. However, for a study of this scope, it would be logical to present similar cross validations of at least 2 to 3 additional identified TF candidates among the 3 subtypes RCC.

Figure 4: allelic imbalance was validated in only one SNP, where the altered C-allele of rs4903064 SNP that created a HIF binding motif was validated in Figures 4E and 4F. Again, similar validations need to be conducted in a few additional identified SNPs as described in Figures 4I-4N.

Page 12, Lines 261-262: "This enrichment represents approximately twice that of, which are themselves enriched 8.1-fold". It was not clear what was "This enrichment" referred to 43.2-fold or 16.7-fold (Page 11, Line 258)?

Finally, the authors could discuss more on clinical relevance and/or application of the study.

REVIEWER COMMENTS

Reviewer #1, expert in ATAC-seq and ChIP-seq (Remarks to the Author):

This manuscript has collected a rich datasets characterizing epigenomic landscapes of clear cell, papillary, and chromophobe RCC cancer cells by using ChIP-seq, ATAC-Seq, RNA-seq, and SNP arrays. They have shown the subtype-specific enhancer regulation profile, predicted master TFs and validated a germline SNP that recognizes HIF-2a with different binding affinity. Although the comprehensive epigenomic atlas in primary patient samples undoubtedly provide a valuable resource for the research community, it is descriptive to me. There are significantly incremental concerns about the uncertain quality control, data interpretation and lack of functional validation, which dampened my enthusiasm to recommend for publication at the current status. I hope that the authors will find the enclosed comments helpful.

We thank the reviewer for acknowledging the value of the data presented in our manuscript and thank them for their excellent input to strengthen our work.

Major comments

1. There are no normal tissues or non-tumor tissues collected accompanying with tumor harvest. Therefore, when using the term of “tumor-specific”, actually there are many H3K27ac marked enhancers that are conserved in normal developmental tissues. Those “subtype-specific” epigenetic signatures, GSEA and pathway analysis should be re-justified by comparing with normal kidney tissue data. I am sure here are plenty datasets available as normal tissue controls at GEO, GTEX or other resources. This is also related to the justification of using one single cell line 786-O for validating the SNP. The authors claimed that this is a typical ccRCC cell line. Did they characterized the epigenomic landscape of this cell line to cluster it with ccRCC with more correlation? I understand the difficulty to translate primary patients samples to cell line, but It is important to choose the right cell type sharing the same epigenomic signature, which is the key point of the this manuscript.

We thank the reviewer for their insightful feedback. We modified the terminology throughout the manuscript from “tumor-specific” to “histology-specific” as we agree this is a more accurate representation of our dataset and analyses.

We have incorporated H3K27ac ChIP-Seq data from 10 normal human kidney samples¹ that were generated in our lab into the results section. We show that the normal kidney samples cluster independently from the tumor tissue independent of the tumor histology supporting the notion that the regulatory landscape is distinct from that of kidney tumors across the inspected histologies. This now reads as “Moreover, unsupervised hierarchical clustering of our cohort with an independent cohort of matched normal (n=10) from the KIRC TCGA cohort¹ showed a clean separation of normal tissue from the different tumor histotypes (chRCC, ccRCC, pRCC, **Figure S2B, S2C**).”

D

Regarding the point behind using 786-O for validating the SNP, we are not entirely sure what the reviewer is referring to. We think this refers to our chromatin allelic imbalance analysis and would like to kindly clarify that the allelic imbalance was performed on human ccRCC and pRCC samples, rather than the cell line (786-O).

In terms of the rationale for using 786-O as the representative cell line, 786-O has been extensively used as the representative cell line for ccRCC and has been cited in many seminal ccRCC papers²⁻⁵.

Nonetheless, we do know that cell lines diverge from primary tissue in vitro, and this particularly why we consider our work important to the scientific community as it provides data on primary human samples.

2. Although multi-Omics data were collected, insufficient integrative analysis as actually conducted. I strongly recommend more validation analysis by using different datasets. For instance, motif enrichment analysis using H3K27ac marked peaks is not powerful strategy given most of the histone modification markers bound broad peaks, great diluting the statistic powder of motif enrichment. However, ChIP-seq of TFs and ATAC-seq peaks were sharp and also correlated with H3K27ac. Since several TFs were predicted, the authors should carry out ATAC-seq motif enrichment analysis as an independent approach to validate the predicted motifs from H3K27ac analysis. Additionally, ATAC-seq with high sequencing resolution can provide footprint signature to support the direct binding. Similarly, CRISPR dropout screens have been extensively used to identify dependency genes and master TFs in kidney cancer. Depmap data should be explored to further support the functional value of the “master TFs”. These integrative analysis should further strengthen the data mining and better support the main conclusion.

We thank the reviewer for this excellent comment. We note that the limitation with only using ATAC-seq for motif enrichment analysis is that the sites could be active or inactive and hence why we initially reported the H3K27ac sites subsetted by ATAC peaks. We carried ATAC-seq motif enrichment analysis as an independent approach to validate the predicted motifs from H3K27ac analysis. We initially set out to analyze histology-specific peaks with at least \log_2 fold change >3 . However, normalization was challenging as we noted a wide variation in ATAC-seq peak densities between the ATAC-seq data we generated on the chrRCC and the published ccRCC and pRCC ATAC-seq dataset⁶. Therefore, we opted to leave this analysis out of the manuscript given technical concerns with the analysis itself.

We agree with the reviewer that dependency has functional implications, however dependency is not synonymous with the function of a master TF. Our work is not trying to claim that candidate TFs are cellular dependencies, but rather that they shape and define cellular identity and the transcriptional landscape. We also note that there are no chromophobe cell lines in DepMap. Using DepMap data with all available RCC cell lines (all ccRCC cell lines), out of 50 candidate master TFs, six TFs (EPAS1, JUNB, MAF, ZEB2, HNF1B, and ESRRA) were identified as partial dependencies in at least one ccRCC cell line (as shown below). We opted to leave the DepMap analyses out of the manuscript as we felt this is beyond the scope of this work.

3. The authors claimed to predict 50 top master TFs which out any other data support how they are controlling the subtype-identify and transcription. The binding occupancy switch at least should be tested in cell lines from different subtypes.

There are unfortunately no chromophobe RCC cell lines publicly available and thus testing the binding occupancy on RCC cell lines is challenging.

We have thus performed ChIP-Seq using the candidate master TF, EPAS1, on two chromophobe human RCC samples as well as four additional clear cell RCC samples and showed differential binding of EPAS1 in chromophobe and clear cell RCCs. We show that there were 2916 clear cell RCC-enriched sites and these elements were enriched for immune cell and white blood cell activation pathways. We also identified 4564 chromophobe RCC-enriched sites that were enriched for metabolic processes and fatty acid activation. We also demonstrate that subtype-specific EPAS1 binding sites are highly enriched for subtype-specific sites of H3K27ac. For instance, 2,090 of the 4,565 chromophobe-specific EPAS1 binding sites (46%) overlapped with chromophobe-specific H3K27ac sites ($p < 2.2e-16$). This supports the

conclusion that these differential EPAS1 binding sites are non-random and biologically relevant, because they coincide with regulatory elements that segregate closely with histology. These results have now been incorporated in the results section of the manuscript and read as:

Results: "We next confirmed histology specific in-vivo binding of the nominated master TF EPAS1 through examining the EPAS1 cistrome in ccRCC and chRCC. We characterized 2916 clear cell RCC-specific and 4564 chromophobe RCC-specific EPAS1 binding sites through performing EPAS1 ChIP on six additional primary human tumors (n = 2 chRCC; n=6 ccRCC, **Figure 3D**). Subtype-specific EPAS1 binding sites were highly enriched for subtype-specific sites of H3K27ac. For instance, 2,090 of the 4,565 chromophobe-specific EPAS1 binding sites (46%) overlapped with chromophobe-specific H3K27ac sites ($p < 2.2e-16$). This supports the conclusion that these differential EPAS1 binding sites are non-random and biologically relevant, because they coincide with regulatory elements that segregate closely with histology. ccRCC-specific EPAS1-binding sites were enriched for immune cell and white blood cell activation pathways and chRCC-specific EPAS1-binding sites were enriched for metabolic processes and fatty acid activation (**Figures 3F, 3G**)."

Discussion:

"Using EPAS1 TF ChIP-Seq as an example, we demonstrate that histology-specific binding of EPAS1 mediates independent pathways in different RCC histologies."

4. Figure 4 starts with ambition but ends of lack of support from GWAS study and functional validation again. The authors should incorporate the previous GWAS hits from large cohort studies with the functional noncoding loci marked by H3K27ac and allele-specific expression (ASE) to further extend the power of epigenomic landscape. Whether the imbalanced binding of TFs lead to ASE of target genes from RNA-seq?

We appreciate the reviewer raising this important point. We wanted to clarify that the GWAS data used in this manuscript is already the largest available cohort study of RCC. The limitation here is that the data available is just clear-cell RCC. In the revised version, we leveraged chromatin AI from our ccRCC H3K27Ac dataset and allele-specific expression (ASE) using RNA-seq data from the KIRC TCGA cohort. SNP loci with at least 50 reads in both the ccRCC H3K27Ac dataset and RNA-seq KIRC TCGA samples were retained. Retained SNPs were classified as chromatin allelically imbalanced or allelically balanced in ccRCC. Chromatin allelically imbalanced SNPs harbored H3K27ac peaks with significant skew towards the alternate allele compared to the wild-type allele ($p < 0.05$). For the 22162 chromatin allelicly imbalanced SNPs, we matched a background set of chromatin allelically balanced SNPs lying within the H3K27ac consensus peak set. Allele-specific expression (ASE) of genes whose transcription start site lies within 50 Kb of the chromatin allelically imbalanced and balanced SNPs was analyzed in the TCGA KIRC dataset (N=412) with significance cut-off of p-adjusted < 0.01 . Of the unique genes analyzed, chromatin allelically imbalanced SNPs were significantly more likely to lie within 50kb of a gene with ASE (1170/2646, 44%) compared the background set of chromatin allelically balanced SNPs (65/940, 6.9%, $p = 8.4e^{-25}$, Table S22-25)

This is now included in the results and reads as "TFs can read the genetic code and bind cis-regulatory elements to activate or repress gene expression. We hypothesized that loci with significant chromatin AI are likely harbingers of TF binding and thus are associated with the regulation of gene expression in an allele-specific manner. We leveraged chromatin AI from our ccRCC H3K27Ac dataset and allele-specific expression (ASE) using RNA-seq data from the KIRC TCGA cohort. SNP loci with at least 50 reads in both the ccRCC H3K27Ac dataset and RNA-seq KIRC TCGA samples were retained. Retained SNPs were classified as chromatin allelically imbalanced or allelically balanced in ccRCC. Chromatin allelically imbalanced SNPs harbored H3K27ac peaks with significant skew towards the alternate allele compared to the wild-type allele ($p < 0.05$). For the 22162 chromatin allelically imbalanced SNPs, we matched a background set of chromatin allelically balanced SNPs lying within the H3K27ac consensus peak set. Allele-specific expression of genes whose transcription start site lies within 50 Kb of the chromatin allelically imbalanced and balanced SNPs was analyzed in the TCGA KIRC dataset (N=412) with significance cut-off of p-adjusted < 0.01 . Of the unique genes analyzed, chromatin allelically imbalanced

SNPs were significantly more likely to lie within 50kb of a gene with ASE (1170/2646, 44%) compared the background set of chromatin allelically balanced SNPs (65/940, 6.9%, $p=8.4e^{-25}$, Table S22-25)”

5. The entire validation of a single SNP at DPF3 locus relies on binding difference. The most direct evidence will be conducting genome editing of either risk or non-risk allele in a relevant cell lines and examine the impact on transcription.

We agree with the reviewer that the most direct evidence comes from genome editing and assessing the impact of perturbing the DPF3 locus on gene expression; however, it is beyond the scope of this manuscript and has been validated in recent work (published while our work was under revision). This is now included in the discussion and reads “Recent work, around the rs4903064 locus⁷ showed that it had an allele-specific effect on DPF3 expression in ACHN and HEK293T cell lines as assessed by massively parallel reporter assay, confirmatory luciferase assays, and eQTL analyses. The rs4903064-C RCC risk allele was shown to create a HIF-binding site and enhance gene expression. Increased expression of DPF3 conferred a growth advantage to cells by at least two pathways: inhibition of apoptosis via CEMIP and activation of STAT3 via IL23R. The authors also showed that DPF3-overexpressing cells showed higher T-cell mediated cytotoxicity compared to controls. In a separate effort, Protze et al.,⁸ used 23 tumor tissue specimens and two primary ccRCC cell lines and showed that the risk SNP is located within an active enhancer region, in turn creating a novel EPAS1 binding motif. They also showed that HIF-mediated DPF3 regulation depends on the presence of the C-risk allele. Moreover, DPF3 deletion in proximal tubular cells decreased cell growth, suggesting a role for DPF3 in cell proliferation.”

6. Quality control (QC) of each datasets should be provided in supplementary figures to show the reproducibility and stats.

We thank the reviewer for this suggestion and have added quality control data for all datasets. This is now included in the supplementary material as Supplementary Table S26. We also included the quality control data in the Methods section which now read as “MACS v2.1.1.20140616⁹ was used for ChIP-seq peak calling with a q-value (FDR) threshold of 0.01. ChIP-seq data quality was evaluated by a variety of measures, including total peak number, FrIP (fraction of reads in peak) score, number of high-confidence peaks (enriched > ten-fold over background), and percent of peak overlap with DHS peaks derived from the ENCODE project”.

7. Figure fonts and character styles are all different. Some of the fonts are too small for me to visualize.

We modified the fonts and character styles throughout the manuscript and figures/tables.

Reviewer #2, expert in renal cell carcinoma genomics/subtypes (Remarks to the Author):

In this paper, Nassar et al. reported an interesting data set of H3K27ac ChIP-seq, H3K4me2 ChIP-seq, ATAC-Seq, RNA-seq, and SNP arrays on human renal cell carcinoma. Through integrated analysis, they identified 50 candidate histology-specific master transcription factors in three main RCC subtypes (ccRCC, pRCC and chRCC). They further evaluated and validated two transcription factors in 786-O cells. By assessing ChIP-seq data, they also annotated risk SNPs identified by a GWAS for RCC. This manuscript contains an abundance of epigenomic profiling data from patient samples and represents an important resource for the field. The figures are generally well prepared. However, I have several concerns about this current version of the paper.

We thank this reviewer for their insightful feedback and appreciation of the contribution of our manuscript to the field.

Major concerns:

1. In contrast to ccRCC and chRCC, a relatively few pRCC samples were performed for ChIP-seq and ATAC-Seq in this study. But pRCC have two main histologic types, type 1 and type 2, which were shown to be different types of renal cancer characterized by specific genetic alterations and biologic behavior. The specific types of pRCC in cohort 1 should be described in detail. If these pRCC all or mainly belong to one pRCC type, it is hard to identify pRCC-specific master transcription factors based on these samples in this study. And more samples of the other type should be added. Also, the analysis about pRCC should be divided into type 1 and type 2.

We appreciate the author's suggestion. We agree that Type 1 and Type 2 pRCC are two distinctive entities. We have now reclassified our cohort of pRCC into type 1 and type 2 after revision of the samples by a certified pathologist. There are 4 type 1 pRCC and 1 type 2 pRCC. For one pRCC, our pathologists were unable to retrieve the H&E slide for revision. This is now included in the results section and reads as "Of the 6 pRCC tumors, 4 were type I, 1 was type II, and 1 was unknown" and the relevant samples in Supplementary table S1 were updated.

Given the small number of samples and the overall similar appearance on unsupervised clustering, we opted to keep the analysis in its original form as it is less likely to find any meaningful difference upon separation into types 1 and 2. We also highlighted this as a limitation in the discussion section which now reads, as:

"It is important to highlight that type 1 and type 2 papillary RCC are distinctive entities, and that although our work can globally highlight differences across RCC histotypes, the small sample size of the pRCC cohort limited more granular assessment by pRCC types."

2. Through integrated analysis, the author identified 50 candidate histology-specific master TFs in three main RCC subtypes, but the expression of these TFs in patient tissue samples need to be validated through IHC or immunofluorescence staining, at least for selected TFs. Furthermore, the potential regulatory circuitries in driving different RCC histologic subtypes should be explored. For example, (1) the binding profiles of these TFs in each RCC subtype; (2) co-regulated relationship; (3) TF hotspots; (4) co-regulated SE domains and the key regulators in the regulatory network.

We thank the reviewer for their excellent feedback and suggestions. We performed IHC on four representative master TFs in up to 10 samples total (three chromophobe RCC samples, three papillary RCC samples, and four clear cell RCC samples). We have now incorporated these findings under the results and discussion sections as well as Figure 3C, included below for reference. We show that expression of BHLHE41, NKX6.1, and HNF1B are specific to ccRCC, chRCC and pRCC respectively. We also show that ZNF395, a ccRCC-specific master TF, is highly expressed in ccRCC.

While we agree that exploring the potential regulatory circuitries in driving different RCC histologic subtypes would be important to further interrogate, we believe it may be best fit for future work. In fact, we

have ongoing projects focused on elucidating mechanisms by which EPAS1 drives pathogenesis across different states of ccRCC (normal, localized ccRCC, metastatic).

We have now included the following under the respective sections in the manuscript:

“Results:

To provide proof of concept validation at the protein level, we investigated the expression specificity and localization of master TFs by immunohistochemistry (IHC). We selected four representative master TFs that met the following criteria: 1) at least 2-fold change in gene expression of the histology-specific TF by bulk RNA-sequencing (from TCGA) compared to the two other RCC histotypes, 2) high-quality antibodies for IHC and 3) TF was not previously validated and implemented on a clinical level. For the 4 TFs (BHLHE41, HNF1B, NKX6.1, and ZNF395), we found significant changes in protein expression levels across histologic subtypes (Figure 3C). More specifically, two ccRCC-specific master TFs (BHLHE41 and ZNF395), were highly expressed in ccRCC tumors.

Both TFs had nuclear localization in 4 out of 4 and 3 out of 3 ccRCC samples, respectively. NKX6.1, a TF recently described as being expressed in chRCC, was detected nuclearly in 2 out of 2 chromophobe samples with no expression in either ccRCC or pRCC.”

Discussion:

“In routine practice, pathologists use a myriad of targets to differentiate across RCC histotypes. Our approach has the potential to narrow master TFs to clinically meaningful ones and augments the current armamentarium with additional TFs including BHLHE41 (ccRCC) and NKX6.1 (chRCC) that can be stained to better characterize tumors.

BHLHE41 was previously shown in triple-negative breast cancer to counteract expression of HIF-target genes by promoting HIF proteasomal degradation in a process independent of VHL or hypoxia¹⁰.

Elevated protein expression of BHLHE41 in ccRCC has not been described previously. Using a systematic approach, we found histology-specific protein expression of BHLHE41 in ccRCC compared to pRCC and chRCC. In ccRCC where HIF activation is foundational, further investigation of the relation between BHLHE41 and HIF activation is warranted.”

Methods:

For immunohistochemistry (IHC), 5mm FFPE tissue sections were deparaffinized, rehydrated, and subjected to heat-induced antigen retrieval (0.01 mol/L sodium citrate tribasic dihydrate, pH 6.0 or EDTA retrieval buffer, pH 9.0) followed by treatment with 3% H₂O₂ for 15 minutes at room temperature to block endogenous peroxidase activity. Sections were incubated overnight at 4°C with primary antibodies against mouse HNF-1B (Santa Cruz, sc-130407, 1:50 dilution), mouse BHLHE41 (ThermoFisher, TA806146, 1:200 dilution), rabbit NKX6.1 (Cell signaling, #54551, 1:50 dilution), rabbit ZNF395 (Lsbio, LS-B5647-100, 1:500 dilution). Sections were then stained with appropriate secondary antibodies and antigen detection was done with ImmPRESS HRP anti-rabbit and anti-mouse IgG Polymer detection kits (Vector MP-7451; MP-7402). Slides were counterstained with hematoxylin, dehydrated, and mounted. Slides were scanned at 20x magnification and positively stained tumor cells for HNF-1B, BHLHE41, NKX6.1, or ZNF395 were determined.”

3. Are the histology-specific master TFs identified in this study associated with patient clinical features and prognosis?

We thank the reviewer for their insightful question. We have looked at the correlation between the expression of the candidate ccRCC-specific master TFs and clinical outcomes using immune checkpoint inhibitor trial data.

We performed gene-by-gene analysis using 311 ccRCC samples from CM009/010/025 trials for all 50 candidate master TFs looking at OS and PFS. After FDR correction (Benjamini-Hochberg <0.1), higher *BARX2* expression was associated with longer OS in the nivo arm and the overall cohort. All other TFs were not associated with survival in the CheckMate cohorts. This is now included in the results section "Clinical correlative analyses from CheckMate cohorts (009/10/025) showed that among the 50 master TFs, high *BARX2* expression significantly correlated with improved overall survival in the entire cohort of patients with ccRCC (Table S13, Figure S7E) and in the subgroup treated with the anti-PD1, nivolumab (Table S13, Figures S7F)."

This however does not necessarily affect our original hypothesis that these proposed master TFs are related to histologic identity rather than aggressiveness and disease prognosis.

All Patients Genes	Overall Survival (OS)					Progression-Free Survival (PFS)				
	HR	Lower CI	Higher CI	p val	q val	HR	Lower CI	Higher CI	p val	q val
AHR	0.95	0.79	1.16	0.64	0.85	0.98	0.83	1.16	0.79	1.00
ARNT2	1.05	0.95	1.15	0.35	0.66	1.04	0.95	1.13	0.42	0.88
BARX2	0.92	0.87	0.96	0.0008	0.0207	0.96	0.91	1.01	0.11	0.71
BCL6	1.00	0.84	1.18	0.97	1.00	1.06	0.91	1.24	0.42	0.88
BHLHE41	0.94	0.84	1.05	0.29	0.66	0.94	0.84	1.05	0.26	0.78
DDIT3	1.04	0.91	1.20	0.56	0.84	1.06	0.94	1.19	0.35	0.88

DMRT2	1.02	0.98	1.06	0.44	0.74	0.97	0.93	1.00	0.08	0.71
ELF3	0.97	0.88	1.06	0.49	0.78	1.01	0.94	1.09	0.80	1.00
EPAS1	0.90	0.77	1.05	0.18	0.58	0.99	0.86	1.14	0.88	1.00
ESRRA	1.00	0.88	1.13	0.98	1.00	1.09	0.97	1.21	0.14	0.71
ETS1	0.91	0.75	1.10	0.33	0.66	0.99	0.83	1.18	0.95	1.00
FOXC1	1.12	0.99	1.26	0.08	0.41	1.04	0.93	1.16	0.48	0.88
FOXI1	1.00	0.96	1.04	0.93	0.99	0.99	0.96	1.03	0.56	0.88
FOXJ3	1.00	0.85	1.17	1.00	1.00	1.04	0.90	1.19	0.61	0.88
FOXP1	1.03	0.87	1.22	0.69	0.85	1.00	0.87	1.15	0.99	1.00
FOXQ1	0.99	0.94	1.04	0.57	0.84	1.02	0.97	1.07	0.51	0.88
GATA2	0.87	0.81	0.94	0.00	0.02	0.93	0.87	0.99	0.03	0.59
HOXA10	1.07	0.98	1.16	0.11	0.45	1.03	0.96	1.11	0.40	0.88
HOXA3	1.05	0.95	1.15	0.35	0.66	1.00	0.92	1.09	0.96	1.00
HOXA7	0.98	0.94	1.03	0.48	0.78	1.01	0.97	1.06	0.59	0.88
ID1	1.01	0.91	1.11	0.88	0.96	1.00	0.92	1.09	0.98	1.00
IRF1	1.09	0.92	1.30	0.31	0.66	0.93	0.80	1.09	0.37	0.88
IRX3	0.98	0.90	1.07	0.67	0.85	1.10	1.00	1.20	0.05	0.59
JUNB	0.97	0.85	1.11	0.67	0.85	0.96	0.86	1.08	0.53	0.88
MAF	0.82	0.71	0.96	0.01	0.11	0.98	0.86	1.13	0.82	1.00
MBNL2	1.03	0.85	1.24	0.75	0.85	1.04	0.88	1.23	0.66	0.92
MECOM	0.87	0.77	0.98	0.03	0.20	0.92	0.83	1.03	0.15	0.71
MXI1	0.90	0.76	1.07	0.22	0.64	1.05	0.90	1.22	0.55	0.88
NKX6-1	1.06	1.00	1.13	0.07	0.41	1.02	0.96	1.08	0.60	0.88
NR1H4	0.93	0.87	1.00	0.05	0.37	0.98	0.92	1.05	0.61	0.88
NR2F2	1.05	0.88	1.25	0.58	0.84	1.01	0.87	1.18	0.87	1.00
NR3C1	0.90	0.72	1.13	0.36	0.66	0.88	0.71	1.08	0.22	0.77
NR4A1	0.95	0.87	1.03	0.20	0.60	0.95	0.89	1.02	0.20	0.75
PAX2	0.97	0.92	1.03	0.35	0.66	1.03	0.98	1.08	0.24	0.77
PRDM4	1.05	0.81	1.36	0.71	0.85	1.00	0.80	1.26	0.97	1.00
SMAD3	0.87	0.71	1.06	0.16	0.56	1.06	0.88	1.27	0.54	0.88
SOX4	1.02	0.90	1.16	0.76	0.85	1.13	1.00	1.28	0.05	0.59
SOX9	1.04	0.95	1.15	0.39	0.68	1.03	0.95	1.13	0.48	0.88
TBX2	0.85	0.76	0.95	0.00	0.07	1.00	0.91	1.10	1.00	1.00
TEF	0.92	0.86	0.98	0.01	0.11	1.01	0.94	1.08	0.87	1.00
TFCP2L1	1.04	0.99	1.10	0.11	0.45	1.02	0.98	1.06	0.41	0.88
TGIF1	1.13	0.91	1.41	0.26	0.66	1.14	0.93	1.38	0.20	0.75
THRB	0.94	0.86	1.02	0.13	0.51	0.96	0.89	1.04	0.32	0.88
TSC22D3	1.05	0.95	1.17	0.33	0.66	1.07	0.97	1.18	0.15	0.71
VDR	0.99	0.91	1.07	0.77	0.85	0.95	0.88	1.02	0.16	0.71
ZBTB7B	0.97	0.80	1.16	0.72	0.85	1.15	0.97	1.37	0.12	0.71
ZEB2	0.86	0.67	1.12	0.27	0.66	0.78	0.63	0.98	0.03	0.59
ZNF395	0.91	0.83	1.01	0.08	0.41	1.00	0.91	1.11	0.98	1.00
ZNF503	1.03	0.93	1.14	0.60	0.84	1.02	0.92	1.13	0.73	1.00

Nivolumab Arm	Overall Survival (OS)	Progression-Free Survival (PFS)
---------------	-----------------------	---------------------------------

Genes	HR	Lower CI	Higher CI	p val	q val	HR	Lower CI	Higher CI	p val	q val
AHR	0.96	0.74	1.24	0.77	0.85	0.98	0.79	1.21	0.87	0.96
ARNT2	1.09	0.96	1.25	0.18	0.47	1.06	0.95	1.18	0.28	0.71
BARX2	0.89	0.83	0.95	0.00048	0.023	0.96	0.89	1.02	0.19	0.59
BCL6	0.98	0.80	1.21	0.87	0.91	0.99	0.82	1.20	0.94	0.96
BHLHE41	0.94	0.80	1.10	0.43	0.69	0.95	0.83	1.09	0.45	0.84
DDIT3	1.08	0.91	1.29	0.39	0.66	1.00	0.87	1.16	0.95	0.96
DMRT2	1.01	0.96	1.06	0.77	0.85	0.96	0.91	1.00	0.08	0.59
ELF3	1.09	0.95	1.25	0.24	0.50	1.06	0.95	1.18	0.30	0.71
EPAS1	0.82	0.67	1.02	0.07	0.31	1.02	0.86	1.22	0.81	0.96
ESRRA	1.05	0.90	1.22	0.52	0.72	1.10	0.97	1.25	0.15	0.59
ETS1	0.75	0.58	0.97	0.03	0.29	0.96	0.76	1.20	0.70	0.90
FOXC1	1.17	0.99	1.37	0.06	0.31	1.10	0.95	1.26	0.19	0.59
FOXI1	0.98	0.93	1.03	0.45	0.69	0.98	0.94	1.03	0.46	0.84
FOXJ3	0.90	0.74	1.09	0.28	0.55	1.01	0.85	1.19	0.95	0.96
FOXP1	1.04	0.85	1.29	0.69	0.82	1.02	0.86	1.21	0.85	0.96
FOXQ1	0.98	0.92	1.05	0.60	0.80	1.03	0.97	1.10	0.30	0.71
GATA2	0.88	0.80	0.97	0.01	0.24	0.95	0.88	1.03	0.20	0.59
HOXA10	1.07	0.96	1.20	0.24	0.50	1.02	0.93	1.12	0.66	0.90
HOXA3	1.13	0.98	1.30	0.08	0.31	1.04	0.93	1.16	0.48	0.84
HOXA7	1.01	0.95	1.08	0.67	0.82	1.05	0.99	1.12	0.08	0.59
ID1	1.01	0.89	1.14	0.87	0.91	1.00	0.90	1.10	0.96	0.96
IRF1	1.18	0.94	1.49	0.15	0.47	0.91	0.74	1.11	0.35	0.78
IRX3	0.96	0.85	1.08	0.45	0.69	1.13	1.01	1.27	0.03	0.59
JUNB	0.89	0.74	1.06	0.20	0.50	0.89	0.76	1.03	0.13	0.59
MAF	0.83	0.69	1.01	0.06	0.31	1.01	0.85	1.20	0.94	0.96
MBNL2	0.97	0.75	1.26	0.83	0.90	1.04	0.83	1.31	0.70	0.90
MECOM	0.82	0.69	0.97	0.02	0.24	0.91	0.80	1.04	0.17	0.59
MXI1	0.83	0.67	1.03	0.09	0.31	1.04	0.86	1.25	0.71	0.90
NKX6-1	1.05	0.98	1.14	0.18	0.47	1.02	0.95	1.09	0.64	0.90
NR1H4	0.92	0.84	1.01	0.09	0.31	0.99	0.91	1.08	0.90	0.96
NR2F2	0.99	0.79	1.24	0.94	0.94	1.05	0.86	1.28	0.63	0.90
NR3C1	0.90	0.68	1.20	0.48	0.71	0.82	0.63	1.06	0.13	0.59
NR4A1	0.89	0.81	0.99	0.04	0.31	0.94	0.86	1.02	0.14	0.59
PAX2	0.97	0.91	1.04	0.39	0.66	1.04	0.98	1.11	0.19	0.59
PRDM4	0.93	0.67	1.28	0.64	0.82	0.95	0.72	1.25	0.72	0.90
SMAD3	0.92	0.72	1.19	0.54	0.74	1.08	0.87	1.35	0.49	0.84
SOX4	1.03	0.87	1.21	0.75	0.85	1.15	0.99	1.33	0.06	0.59
SOX9	1.09	0.96	1.24	0.17	0.47	1.08	0.97	1.20	0.15	0.59
TBX2	0.84	0.74	0.97	0.02	0.24	1.04	0.93	1.17	0.47	0.84
TEF	0.90	0.81	1.01	0.06	0.31	1.07	0.96	1.19	0.20	0.59
TFCP2L1	1.06	0.99	1.12	0.10	0.32	1.01	0.95	1.06	0.82	0.96
TGIF1	1.10	0.83	1.46	0.50	0.71	1.08	0.85	1.38	0.53	0.84
THRB	0.94	0.84	1.05	0.24	0.50	0.96	0.87	1.06	0.45	0.84
TSC22D3	1.03	0.90	1.18	0.69	0.82	1.04	0.92	1.17	0.53	0.84
VDR	1.05	0.94	1.18	0.39	0.66	0.94	0.86	1.03	0.19	0.59
ZBTB7B	0.99	0.78	1.25	0.94	0.94	1.12	0.90	1.40	0.30	0.71
ZEB2	0.83	0.60	1.13	0.24	0.50	0.75	0.58	0.97	0.03	0.59
ZNF395	0.89	0.78	1.01	0.08	0.31	1.05	0.92	1.19	0.48	0.84

ZNF503

1.06 0.93 1.20 0.39 0.66 | 1.03 0.91 1.17 0.60 0.90

Everolimus Arm Genes	Overall Survival (OS)					Progression-Free Survival (PFS)				
	HR	Lower CI	Higher CI	p val	q val	HR	Lower CI	Higher CI	p val	q val
AHR	0.96	0.72	1.28	0.78	0.98	0.98	0.74	1.30	0.89	0.95
ARNT2	0.99	0.86	1.16	0.94	0.98	0.94	0.80	1.10	0.45	0.89
BARX2	0.96	0.88	1.04	0.28	0.80	0.95	0.88	1.03	0.23	0.80
BCL6	1.04	0.78	1.39	0.78	0.98	1.33	1.01	1.74	0.04	0.29
BHLHE41	0.95	0.81	1.12	0.54	0.94	0.91	0.76	1.10	0.33	0.84
DDIT3	1.00	0.80	1.26	0.98	0.98	1.30	1.06	1.59	0.01	0.29
DMRT2	1.03	0.97	1.10	0.32	0.80	0.99	0.93	1.05	0.74	0.95
ELF3	0.86	0.77	0.97	0.02	0.37	0.95	0.86	1.05	0.34	0.84
EPAS1	0.99	0.78	1.25	0.91	0.98	0.86	0.68	1.09	0.21	0.79
ESRRA	0.88	0.70	1.11	0.30	0.80	1.02	0.81	1.30	0.84	0.95
ETS1	1.19	0.88	1.61	0.26	0.79	1.04	0.80	1.35	0.78	0.95
FOXC1	1.01	0.82	1.23	0.95	0.98	0.93	0.79	1.10	0.38	0.84
FOXI1	1.03	0.97	1.09	0.40	0.86	1.01	0.95	1.07	0.76	0.95
FOXJ3	1.19	0.90	1.56	0.22	0.79	1.11	0.85	1.44	0.46	0.89
FOXP1	0.99	0.73	1.35	0.95	0.98	0.93	0.72	1.20	0.60	0.94
FOXQ1	1.01	0.93	1.09	0.85	0.98	0.98	0.91	1.06	0.68	0.95
GATA2	0.83	0.73	0.94	0.00	0.11	0.84	0.75	0.95	0.00	0.20
HOXA10	1.08	0.95	1.23	0.23	0.79	1.06	0.94	1.19	0.33	0.84
HOXA3	0.95	0.83	1.10	0.50	0.91	0.92	0.79	1.08	0.30	0.84
HOXA7	0.93	0.87	1.00	0.06	0.66	0.92	0.86	0.99	0.02	0.29
ID1	1.01	0.83	1.22	0.94	0.98	1.01	0.85	1.21	0.87	0.95
IRF1	0.99	0.76	1.28	0.92	0.98	1.01	0.80	1.27	0.96	0.97
IRX3	1.01	0.88	1.16	0.92	0.98	1.03	0.89	1.18	0.70	0.95
JUNB	1.12	0.92	1.36	0.25	0.79	1.18	0.97	1.42	0.09	0.55
MAF	0.84	0.65	1.08	0.18	0.79	0.92	0.72	1.16	0.47	0.89
MBNL2	1.10	0.84	1.44	0.48	0.91	1.00	0.77	1.28	0.97	0.97
MECOM	0.95	0.78	1.15	0.58	0.94	0.94	0.78	1.14	0.54	0.91
MXI1	1.05	0.80	1.37	0.71	0.98	1.08	0.83	1.39	0.57	0.93
NKX6-1	1.13	0.99	1.28	0.06	0.66	1.02	0.91	1.15	0.75	0.95
NR1H4	0.95	0.86	1.06	0.40	0.86	0.95	0.84	1.07	0.37	0.84
NR2F2	1.07	0.82	1.40	0.61	0.96	0.88	0.67	1.15	0.35	0.84
NR3C1	0.87	0.60	1.26	0.48	0.91	1.13	0.78	1.64	0.51	0.90
NR4A1	1.06	0.93	1.21	0.40	0.86	1.02	0.90	1.16	0.77	0.95
PAX2	0.98	0.90	1.08	0.70	0.98	0.99	0.91	1.08	0.85	0.95
PRDM4	1.37	0.88	2.15	0.16	0.79	1.19	0.79	1.80	0.40	0.85
SMAD3	0.73	0.51	1.04	0.08	0.66	0.96	0.67	1.38	0.84	0.95
SOX4	0.99	0.78	1.27	0.96	0.98	1.03	0.81	1.31	0.83	0.95
SOX9	0.93	0.79	1.11	0.42	0.86	0.84	0.70	0.99	0.04	0.29
TBX2	0.86	0.71	1.03	0.11	0.66	0.80	0.66	0.97	0.02	0.29
TEF	0.93	0.86	1.01	0.10	0.66	0.94	0.87	1.02	0.11	0.55
TFCP2L1	1.01	0.93	1.10	0.82	0.98	1.06	0.98	1.15	0.13	0.56
TGIF1	1.27	0.89	1.81	0.18	0.79	1.34	0.94	1.89	0.10	0.55
THRB	0.91	0.79	1.06	0.24	0.79	0.95	0.83	1.09	0.49	0.89

TSC22D3	1.10	0.92	1.32	0.31	0.80	1.21	1.01	1.45	0.04	0.29
VDR	0.88	0.77	1.01	0.08	0.66	0.98	0.86	1.13	0.80	0.95
ZBTB7B	0.96	0.71	1.31	0.80	0.98	1.25	0.93	1.67	0.14	0.56
ZEB2	0.91	0.56	1.49	0.71	0.98	1.01	0.65	1.56	0.97	0.97
ZNF395	0.97	0.82	1.15	0.71	0.98	0.91	0.78	1.07	0.25	0.83
ZNF503	0.94	0.77	1.15	0.56	0.94	0.99	0.82	1.19	0.89	0.95

Supplementary Figure 7F: Kaplan Meier curve showing median OS in patients treated with Nivolumab stratified by BARX2 expression level.

4. The validation of master TFs should be performed on patient derived organoids or at least primary patient cancer cells rather than 786-O. Furthermore, for the most important finding such as FOXI1 and EPAS1, in vivo validations should be considered.

We agree with the reviewer that further validation of the master TFs (EPAS1 and FOXI1) is warranted, and we hope to perform this in future work using in vivo mouse experiments. However, we believe our work perturbing FOXI1 and EPAS1 serves as a proof-of-concept and provides confidence for future researchers to perturb other candidate TFs we had nominated to study their effect on defining cell identity.

5. After overexpression or suppression the master TFs, only gene expression changes were evaluated. Further functional experiments such as angiogenesis, cell proliferation and migration assays were also

needed. Moreover, the changes of biomarkers for ccRCC (eg, CAIX) and chRCC (eg, CK7, CD117) after the manipulation of TF expression should also be evaluated.

Compared to the 786-O cell line, the double cell line (EPAS1 KO/FOXI1 OE) had significantly higher expression of the chRCC biomarker CD117. In contrast, the WT 786-O cell line had significantly higher expression of the ccRCC biomarker CAIX compared to the double cell line. CK7 expression was comparable across the WT 786-O and EPAS1 KO/FOXI1 OE cell lines. This is now included in the results and reads as “Supervised analysis showed that CAIX, a downstream target of HIF-1 α , was significantly overexpressed in the WT 786-O cell line compared to EPAS1 KO/FOXI1 OE cell line. In contrast, CD117, a well-known immunohistochemical marker of chRCC¹¹, was overexpressed in the EPAS1 KO/FOXI1 OE cell line compared to WT 786-O cell line. Although CK7 expression is a supportive marker clinically for chRCC¹², its expression was uniform across WT 786-O and manipulated cell line states (Figure 4C).”

6. The related results of FOXI1 OE and EPAS1 KD in cells should be added in supplementary figures and tables.

These results are now added to the supplementary figure S9 (referenced below) and tables S15-18.

7. VHL-wild type cells or models should be added in the validation of master TFs for ccRCC.

We appreciate the reviewer's suggestion. VHL is invariably mutated or silenced in patients with ccRCC tumors. As such, we preferred focusing on VHL-mutated cell lines that better reflect the tumor counterparts.

8. Master TFs for pRCC (eg, HNF1B) also need to be validated, rather than discussed it in the discussion section.

We agree with the reviewer that validation of pRCC-associated master TFs is of value. As shown above, we have now validated HNF1 β as a pRCC specific TF using IHC.

We hope that our current work serves as a proof-of-concept that integrative analyses can help narrow down candidate targets. We perturbed two master TFs in a ccRCC cell line and showed that with this, we were able to induce a transcriptional response that partially resembled a chRCC profile. We hope that

future work will utilize this methodology not only to validate other master TFs nominated in this study but also to nominate master TFs in different cancer types. This is now stated as a limitation in the manuscript and reads as “First, experimental validation was limited and performed on two master TFs in one ccRCC cell line 786-O. Future work focused on validation of the role of other candidate master TFs in lineage plasticity and across multiple RCC cell lines is warranted.”

Minor concerns:

1. Fig 2D, HIF-1a should be changed to HIF-2a.

This has been fixed accordingly.

2. Page 9, line 200-201, “FOXI1 scored as a chromophobe-specific TF in 3/4 master TF analyses (Figure 3B).” Figure 3B fails to reflect this result.

We thank the reviewer for this comment. As the reviewer can see below from Figure 3B, FOXI1 was a chromophobe-specific TF in clique enrichment score (CES), differential gene expression (GE), CaCTS prediction, and superenhancer (SE) rank analyses.

3. Page 9, line 204-205, “We also manipulated the expression of EPAS1 as a second target as it was highly specific for ccRCC in our integrative analysis.” This result is shown by Figure S7B.

We have edited the text to reflect this. This now reads as “We also manipulated the expression of *EPAS1* as a second target as it was highly specific for ccRCC in our integrative analysis (Figure 3B, S8B), and prior studies have characterized its role in the pathogenesis of ccRCC¹³.”

4. Page 10, line 213-214, “Of note, the FOXI1 OE/EPAS1 KD cell line did not have significant expression differences compared to FOXI1 OE only.” Perhaps I am missing it, I could not find this result.

We thank the reviewer for this suggestion. It is now included as a supplementary figure and table and reads as “Of note, there were only 109 differentially expressed genes between the FOXI1 OE/EPAS1 KD cell and FOXI1 OE only cell line ($p < 0.05$, $q < 0.01$, Figure S9D, Table S17).”

5. Figure S7, the expression changes of FOXI1 and EPAS1 protein should be added.

We thank the reviewer for this suggestion. We added the western blotting for FOXI1 OE and EPAS1 KD to figure S7E, also added below for reference.

Legend: Western blot, with quantification, to confirm knockdown of EPAS1 (Left) and overexpression of FOXI1 (right) in 786-O cells. Densitometry for EPAS1 and FOXI1 was performed using ImageJ and normalized to ACTB. Sh: short hairpin. NTC: non-targeting control. WT: Wild type. OE: overexpression.

6. Figure S8, Volcano plot of RNA-Seq data showing differentially expressed for 786-O CTRL vs. FOXI1 OE should be added.

We appreciate the reviewer's comment. We have now added the volcano plot of RNA-seq showing differential expression between 786-O control and FOXI1 OE (updated Supplementary figure S9B).

7. Figure S8, figure S8C was missed.
This was updated accordingly in the figure.

8. Figure S8D, the comparative objects should be added.

The comparative objects have now been added to the figure.

9. Page 12, line 273, HIF2-alpha should be changed to HIF2- α .
We have adjusted this accordingly.

Reviewer #3, expert in renal cell carcinoma GWAS (Remarks to the Author):

This is a well-executed comprehensive epigenetic study that has identified 50 candidate histology-specific master transcription factors (TF) among renal cancer carcinoma (RCC). The work is significant to fill the knowledge gap in the area. The study is unique to distinct three histology types of RCC, i.e., clear cell (ccRCC), papillary (pRCC), and chromophobe (chRCC) from 42 patient tumor sample analysis.

We appreciate the reviewer's positive feedback on our manuscript.

Figure 3: the authors cross validated that overexpression of a single chRCC master transcription factor, FOXI1, in ccRCC cell line 786-O, with or without knockdown of EPAS1, led to marked expression changes driving the cell line to be more like a chRCC cell line without any modification to the set of mutations present in 786-O cells. However, for a study of this scope, it would be logical to present similar cross validations of at least 2 to 3 additional identified TF candidates among the 3 subtypes RCC.

We agree with the reviewer that it is important to validate additional candidate key transcription factors proposed by this study. Our work was a proof of concept in a clear cell RCC cell line. We are, however, limited as there are no available chromophobe cell lines. We tried to address this limitation by performing IHC on the candidate master TFs as shown below. We have also added this to our limitations section.

“Results:

To provide proof of concept validation at the protein level, we investigated the expression specificity and localization of master TFs by immunohistochemistry (IHC). We selected four representative master TFs that met the following criteria: 1) at least 2-fold change in gene expression of the histology-specific TF by bulk RNA-sequencing (from TCGA) compared to the two other RCC histotypes, 2) high-quality antibodies for IHC and 3) TF was not previously validated and implemented on a clinical level. For the 4 TFs (BHLHE41, HNF1B, NKX6.1, and ZNF395), we found significant changes in protein expression levels across histologic subtypes (Figure 3C). More specifically, two ccRCC-specific master TFs (BHLHE41 and ZNF395), were highly expressed in ccRCC tumors.

Both TFs had nuclear localization in 4 out of 4 and 3 out of 3 ccRCC samples, respectively. NKX6.1, a TF recently described as being expressed in chRCC, was detected nuclearly in 2 out of 2 chromophobe samples with no expression in either ccRCC or pRCC.

Discussion:

In routine practice, pathologists use a myriad of targets to differentiate across RCC histotypes. Our approach has the potential to narrow master TFs to clinically meaningful ones and augments the current armamentarium with additional TFs including BHLHE41 (ccRCC) and NKX6.1 (chRCC) that can be stained to better characterize tumors.

BHLHE41 was previously shown in triple-negative breast cancer to counteract expression of HIF-target genes by promoting HIF proteasomal degradation in a process independent of VHL or hypoxia¹⁰. Elevated protein expression of BHLHE41 in ccRCC has not been described previously. Using a systematic approach, we found histology-specific protein expression of BHLHE41 in ccRCC compared to pRCC and chRCC. In ccRCC where HIF activation is foundational, further investigation of the relation between BHLHE41 and HIF activation is warranted.

Limitations:

First, experimental validation was limited and performed on two master TFs in one ccRCC cell line 786-O. Future work focused on validation of the role of other candidate master TFs in lineage plasticity and across multiple RCC cell lines is warranted.

Methods:

For immunohistochemistry (IHC), 5mm FFPE tissue sections were deparaffinized, rehydrated, and subjected to heat-induced antigen retrieval (0.01 mol/L sodium citrate tribasic dihydrate, pH 6.0 or EDTA retrieval buffer, pH 9.0) followed by treatment with 3% H₂O₂ for 15 minutes at room temperature to block endogenous peroxidase activity. Sections were incubated overnight at 4°C with primary antibodies

against mouse HNF-1B (Santa Cruz, sc-130407, 1:50 dilution), mouse BHLHE41 (Thermofisher, TA806146, 1:200 dilution), rabbit NKX6.1 (Cell signaling, #54551, 1:50 dilution), rabbit ZNF395 (Lsbio, LS-B5647-100, 1:500 dilution). Sections were then stained with appropriate secondary antibodies and antigen detection was done with ImmPRESS HRP anti-rabbit and anti-mouse IgG Polymer detection kits (Vector MP-7451; MP-7402). Slides were counterstained with hematoxylin, dehydrated, and mounted. Slides were scanned at 20x magnification and positively stained tumor cells for HNF-1B, BHLHE41, NKX6.1, or ZNF395 were determined.”

Figure 4: allelic imbalance was validated in only one SNP, where the altered C-allele of rs4903064 SNP that created a HIF binding motif was validated in Figures 4E and 4F. Again, similar validations need to be conducted in a few additional identified SNPs as described in Figures 4I-4N.

We have looked at HIF2- α /EPAS1 binding as a function of genotype at the SNPs shown in figs 4 I-J. The highlighted SNPs did not seem to affect the HIF2- α /EPAS1 peaks they overlap with. This isn't too surprising, because none of them directly disrupt a HIF2- α /EPAS1 motif (they may be changing peak activity through effects on other TFs, or in linkage disequilibrium with other SNPs that do so). In an independent effort using ATAC-seq data from 406 TCGA samples across 23 cancer types, an integrated analyses incorporating germline allele-specific accessibility QTLs, regulome-wide associations study, and homer TF motif discovery showed that rs7132434 is a causal risk variant in renal cell carcinoma (N=49 RCC samples) but not in any of the other cancer types analyzed (Supplementary Table S8 of the paper)¹⁴.

Page 12, Lines 261-262: "This enrichment represents approximately twice that of . . . , which are themselves enriched 8.1-fold". It was not clear what was "This enrichment" referred to 43.2-fold or 16.7-fold (Page 11, Line 258)?

We have now edited this. This now reads as "This enrichment represents more than five-fold that of the total set of H3K27ac peaks, which are themselves enriched 8.1-fold ($p = 1.2 \times 10^{-4}$)."

Finally, the authors could discuss more on clinical relevance and/or application of the study.

We have expanded on the clinical relevance of this study in the discussion section which now reads as "In routine practice, pathologists use a myriad of targets to differentiate across RCC histotypes. Our approach has the potential to narrow master TFs to clinically meaningful ones and augments the current armamentarium with additional TFs including BHLHE41 (ccRCC) and NKX6.1 (chRCC) that can be stained to better characterize tumors.

BHLHE41 was previously shown in triple-negative breast cancer to counteract expression of HIF-target genes by promoting HIF proteasomal degradation in a process independent of VHL or hypoxia¹⁰.

Elevated protein expression of BHLHE41 in ccRCC has not been described previously. Using a systematic approach, we found histology-specific protein expression of BHLHE41 in ccRCC compared to pRCC and chRCC. In ccRCC where HIF activation is foundational, further investigation of the relation between BHLHE41 and HIF activation is warranted."

"Using clinical trial data (CheckMate 009/010/025), we suggest that high BARX2 expression may be associated with longer survival among patients treated with nivolumab but not everolimus. Prospective data in the metastatic ccRCC space are needed to confirm the role of BARX2 expression as a predictive biomarker in patients treated with immune checkpoint inhibitors."

References:

- 1 Gusev, A. *et al.* Allelic imbalance reveals widespread germline-somatic regulatory differences and prioritizes risk loci in Renal Cell Carcinoma. *bioRxiv*, 631150, doi:10.1101/631150 (2019).
- 2 Cho, H. *et al.* On-target efficacy of a HIF-2alpha antagonist in preclinical kidney cancer models. *Nature* **539**, 107-111, doi:10.1038/nature19795 (2016).
- 3 Stransky, L. A. *et al.* Sensitivity of VHL mutant kidney cancers to HIF2 inhibitors does not require an intact p53 pathway. *Proc Natl Acad Sci U S A* **119**, e2120403119, doi:10.1073/pnas.2120403119 (2022).
- 4 Sinha, R. *et al.* Analysis of renal cancer cell lines from two major resources enables genomics-guided cell line selection. *Nat Commun* **8**, 15165, doi:10.1038/ncomms15165 (2017).
- 5 Yao, X. *et al.* VHL Deficiency Drives Enhancer Activation of Oncogenes in Clear Cell Renal Cell Carcinoma. *Cancer Discov* **7**, 1284-1305, doi:10.1158/2159-8290.CD-17-0375 (2017).
- 6 Corces, M. R. *et al.* The chromatin accessibility landscape of primary human cancers. *Science* **362**, doi:10.1126/science.aav1898 (2018).
- 7 Colli, L. M. *et al.* Altered regulation of DPF3, a member of the SWI/SNF complexes, underlies the 14q24 renal cancer susceptibility locus. *Am J Hum Genet* **108**, 1590-1610, doi:10.1016/j.ajhg.2021.07.009 (2021).
- 8 Protze, J. *et al.* The renal cancer risk allele at 14q24.2 activates a novel hypoxia-inducible transcription factor-binding enhancer of DPF3 expression. *J Biol Chem* **298**, 101699, doi:10.1016/j.jbc.2022.101699 (2022).
- 9 Zhang, Y. *et al.* Model-based analysis of ChIP-Seq (MACS). *Genome Biol* **9**, R137, doi:10.1186/gb-2008-9-9-r137 (2008).
- 10 Montagner, M. *et al.* SHARP1 suppresses breast cancer metastasis by promoting degradation of hypoxia-inducible factors. *Nature* **487**, 380-384, doi:10.1038/nature11207 (2012).
- 11 Liu, L. *et al.* Immunohistochemical analysis of chromophobe renal cell carcinoma, renal oncocytoma, and clear cell carcinoma: an optimal and practical panel for differential diagnosis. *Arch Pathol Lab Med* **131**, 1290-1297, doi:10.5858/2007-131-1290-IAOCRC (2007).
- 12 Mathers, M. E., Pollock, A. M., Marsh, C. & O'Donnell, M. Cytokeratin 7: a useful adjunct in the diagnosis of chromophobe renal cell carcinoma. *Histopathology* **40**, 563-567, doi:10.1046/j.1365-2559.2002.01397.x (2002).
- 13 Shen, C. & Kaelin, W. G., Jr. The VHL/HIF axis in clear cell renal carcinoma. *Semin Cancer Biol* **23**, 18-25, doi:10.1016/j.semcancer.2012.06.001 (2013).
- 14 Grishin, D. & Gusev, A. Allelic imbalance of chromatin accessibility in cancer identifies candidate causal risk variants and their mechanisms. *Nat Genet* **54**, 837-849, doi:10.1038/s41588-022-01075-2 (2022).

Reviewers' Comments:

Reviewer #1:

Remarks to the Author:

The authors have appropriately addressed most of my comments and provided a justification for those they could not do according to technical challenges or disagreement. However, I have to point out that the display of figures with different fonts and word styles is very unpleasant to read. In particular, I could not read any details in Figure 3D with minimal fonts embedded. My question is not just limited to Figure 3D but across the entire figures 1-5 and supplementary materials. The authors claimed that they have carefully modified the fonts and character styles throughout the manuscript and figures/tables, but they didn't. At least they should keep the fonts visible and consistent as a minimal requirement. Therefore, I will leave this issue to editor for consideration.

Reviewer #2:

Remarks to the Author:

I commend the authors on a thorough and excellent response to my comments and concerns. I appreciate the effort that they put into addressing each point, and I have no further concerns. Are the BAM files available for others to reuse as a resource?

Reviewer #3:

Remarks to the Author:

The authors has addressed my comments very well. I have no other concerns.

REVIEWERS' COMMENTS

Reviewer #1 (Remarks to the Author):

The authors have appropriately addressed most of my comments and provided a justification for those they could not do according to technical challenges or disagreement. However, I have to point out that the display of figures with different fonts and word styles is very unpleasant to read. In particular, I could not read any details in Figure 3D with minimal fonts embedded. My question is not just limited to Figure 3D but across the entire figures 1-5 and supplementary materials. The authors claimed that that they have carefully modified the fonts and character styles throughout the manuscript and figures/tables, but they didn't. At least they should keep the fonts visible and consistent as a minimal requirement. Therefore, I will leave this issue to editor for consideration.

We thank the reviewer for their continuous valuable input. We revisited each figure in detail and adjusted the fonts such that the titles are in font 12 Arial, the axes are in font 10 Arial, and the keywords are in font 8 Arial. However, for the heatmaps in figure S5, we were unable to label the samples using the 10 font due to size limitations.

Reviewer #2 (Remarks to the Author):

I commend the authors on a thorough and excellent response to my comments and concerns. I appreciate the effort that they put into addressing each point, and I have no further concerns.

Are the BAM files available for others to reuse as a resource?

We thank the reviewer for their kind words and prior comments that led to significant improvement of our work. Yes, the BAM files for the DFCI datasets generated during and/or analyzed during the current study are available in the GEO repository (GSE188486).

Reviewer #3 (Remarks to the Author):

The authors has addressed my comments very well. I have no other concerns.

We thank the reviewer for their kind words and for their important contributions to the review process.